# Estimating the Empowerment of Language Model Agents

## Abstract

As language model (LM) agents become more capable and gain broader access to real-world tools, there is a growing need for scalable evaluation frameworks of agentic capability. However, conventional benchmark-centric evaluations are costly to design and require human designers to come up with valid tasks that translate into insights about general model capabilities. In this work, we propose information-theoretic evaluation based on *empowerment*, the mutual information between an agent's actions and future states, as an open-ended method for evaluating LM agents. We introduce **EELMA (Estimating Empowerment of Language Model Agents)**, an algorithm for approximating effective empowerment from multi-turn text interactions. We validate EELMA on both language games and scaled-up realistic web-browsing scenarios. We find that empowerment strongly correlates with average task performance, characterize the impact of environmental complexity and agentic factors such as chain-of-thought, model scale, and memory length on estimated empowerment, and that high empowerment states and actions are often pivotal moments for general capabilities. Together, these results demonstrate empowerment as an appealing general-purpose metric for evaluating and monitoring LM agents in complex, open-ended settings. Code available: `https://anonymous.4open.science/r/EELMA-E227`

## 1 Introduction

Large language model agents (LM-agents) are now capable of acting proactively within and across broader computational systems. In this agentic paradigm, LLMs are expected to make autonomous decisions, invoke external tools such as search engines or APIs to access real-time information (Schick et al., 2023), control operating systems and development environments to configure settings (Kwon et al., 2024), and engage in multi-agent interactions with humans or other AIs (Li et al., 2024). However, as these interactions occur over longer time horizons and with greater complexity, evaluating LLM agent performance and safety has become a time-consuming and costly challenge.

Most current evaluations rely on *goal-centric benchmarks* (Phuong et al., 2024; Zhou et al., 2023), where human-designed tasks serve as proxies for capability. While this approach enables direct and practical assessment, it suffers from two limitations. First, designing large-scale evaluation tasks is labor-intensive and challenging. Second, traditional evaluation rarely considers the dynamic and open-ended nature of agentic interactions (Stanley & Lehman, 2015). Instead, the focus is more narrowly on specific end goals or hand-selected milestones. As a result, traditional evaluations are unable to detect when agents are capably pursuing goals outside the measured scope. This blind spot matters for AI safety because it can hide capability growth that benchmarks fail to capture.

To address the gap, we propose leveraging *empowerment*, an information-theoretic measure of an agent's influence (Klyubin et al., 2005; Salge et al., 2014; Myers et al., 2025), to quantify LM agent capability without specifying goals. Consider the agent-environment framework where an LM agent observes its current state (e.g., a webpage or code editor), takes actions (clicking, typing, or generating responses), and transitions to new states (Figure 1). Empowerment quantifies how much control an agent has over future states through its actions. Highly empowered agents recognize the full range of available actions (optionality) and can effectively chain them together to navigate to diverse future states. Therefore, empowerment serves as a strong candidate metric for formalizing a general notion of agentic capability. However, classical empowerment estimators (Klyubin et al., 2005; Jung et al.,

Figure 1: **Empowerment reflects an agent's ability to reach diverse future states.** (Top) A low-empowerment LM-agent becomes trapped in a loop and thus can access only a small fraction of states. (Bottom) A high-empowerment LM-agent effectively explores a wider range of trajectories and can successfully reach states that solve different random goals.

2011) are computationally expensive and do not scale to high-dimensional language-driven settings of LM agents. Tasks such as browsing, coding, or dialogue involve natural language variability, semantic sparsity, and uncertainty in state transitions, which make estimation particularly challenging. This motivates the need for a new scalable estimation algorithm tailored to language-based agents.

In this paper, we propose **EELMA** (Empowerment Estimation for Language Model Agents), a framework that estimates empowerment from multi-turn language interactions. EELMA enables scalable, goal-agnostic measurement of LM-agent capability without explicit task specifications or reward functions. We validate our approach first in structured language games (Gridworld and Tower of Hanoi) and then in a scaled-up web-browsing sandbox (WebArena (Zhou et al., 2023)). In each of these settings, we show that empowerment estimates strongly correlate with average task performance. This highlights the potential of using EELMA for standalone evaluation of an agent's capability, without relying on task-specific reward functions or hand-crafted scores. Furthermore, high-empowerment moments in a trajectory reveals critical points where agents rapidly expand their control over the environment. This makes empowerment not only a promising tool for evaluation but also a potential diagnostic tool for monitoring unintended behavior during training and deployment. Our main contributions are:

1. **First Empowerment Estimator for Text Environments:** We develop a novel information-theoretic estimator, **EELMA** (Empowerment Estimation for Language Model Agents). EELMA is the first method to estimate effective empowerment directly from multi-turn text-based interactions.

2. **Formalization & Validation of Empowerment as Goal-Agnostic Evaluation Metric:** We theoretically and emprically show that effective empowerment (as measured by EELMA) enables goal-agnostic evaluation of LM-agent capability in both toy environments and a scaled-up web-browsing task.

3. **Comprehensive Empowerment Analysis:** We analyze how LM-agent subsystems such as chain-of-thought, memory capacity, and backbone LLM architecture change the effective empowerment of the agent.

4. **Identification of Influential Steps:** Finally, we demonstrate that empowerment highlights highly influential states in a trajectory without human annotation, which may provide a scalable mechanism for open-ended monitoring of anomalous behavior.

## 2 RELATED WORKS

**Large Language Model Agents and Benchmarking** The advancements in Large Language Models (LLMs) have led to a new class of autonomous agents, referred to as LM-agents (Yao et al., 2023; Aksitov et al., 2023; Pan et al., 2024). In these systems, the agent perceives an environment state or context, generates a plan, and executes an action. Multi-turn interactions, often augmented with memory or planning summaries, enable LM-agents to tackle tasks requiring context, long horizons, and complex reasoning (Xu et al., 2025). Benchmarks for agents evaluate their behavior in domains

such as software engineering (Jimenez et al., 2023; Aleithan et al., 2024), web navigation (Zhou et al., 2023), games (Anonymous, 2024), and practical computing (Xie et al., 2024). These benchmarks rely on handcrafted completion or milestone-based goal metrics. In contrast, we quantify an agent's control over the environment using an information-theoretic approach, offering a complementary evaluation methodology.

**Information Theoretic Measures** *Empowerment* is an information theoretic measure that quantifies an agent's ability to influence its environment. Formally, it is defined as the channel capacity between an agent's actions and its subsequent sensory inputs, capturing the maximal mutual information between the agent's actions and future states (Salge et al., 2014). Variational techniques are now available to estimate empowerment in high dimensional, continuous domains (Mohamed & Rezende, 2015). Furthermore, recent work has used the mutual information between actions and states as an intrinsic reward signal for training RL agents to encourage exploration (Bharadhwaj et al., 2022) or assist humans without needing to infer their goals (Myers et al., 2025). In contrast to the above methods, which have been limited to robotic and reinforcement learning tasks, our work enables information-theoretic measurement of influence for LM-agents operating in text environments.

## 3 METHOD: EMPOWERMENT ESTIMATION OF LM AGENT FROM LANGUAGE-BASED MULTITURN TRAJECTORIES

We formalize Language Model (LM) agents within the standard framework of a Markov Decision Process (MDP), represented by the tuple: $(\mathcal{S}, \mathcal{A}, T, R, \gamma)$, where $s \in \mathcal{S}$ denotes the underlying symbolic environment state, $a \in \mathcal{A}$ represents an action executed by the agent. The dynamics are governed by the transition probability function $T(s'|s, a)$, and the rewards (goals) are distributed by the reward function $R(s)$. The discount factor $\gamma$ determines how future rewards are weighted. At each step, given the current state $s$, the LM agent samples an action according to its policy $\pi_{\text{LM}}(a \mid s, P)$, where $P$ denotes the prompt context, including the system prompt, memories, and any Chain-of-Thought (CoT) reasoning.

**Empowerment** Empowerment is an information-theoretic measure of an agent's ability to influence its environment (Klyubin et al., 2005; Myers et al., 2025; Salge et al., 2014). In multi-turn interactions, an empowered agent exerts greater influence on subsequent states. This influence is quantified by the mutual information between the agent's current action and the resulting future state, essentially measuring how decisively the current action determines future outcomes.

We now formally define effective empowerment. To consider the influence of an agent's action on the future, we introduce the random variable $s_*$ representing a future state sampled $\tau \sim \text{Geom}(1 - \gamma)$ steps ahead under the policy $\pi_{LM}$. The agent's control over $s_*$ is then the mutual information between the agent's current action $a_t$ and $s_*$. Formally, $I(a_t; s_* \mid s_t) \triangleq \mathbb{E}_{\tau, s^*, a_t} \left[ \log \frac{P(s_{t+\tau} = s_* \mid s_t, a_t)}{P(s_{t+\tau} = s_* \mid s_t)} \right]$.

Our core metric, effective empowerment $\mathcal{E}$, is defined as the average mutual information between the agent's action $a_t$ and the future state $s^*$ with discounted factor $\gamma$:

$$\mathcal{E}(\pi_{LM}) \triangleq \mathbb{E}_{s_t, a_t, s_*}\left[I(a_t; s_* \mid a)\right] = \mathbb{E}\left[\sum_{t=0}^{\infty} \frac{\gamma^t}{1 - \gamma} \log \frac{P(s_{t+\tau} = s_* \mid s_t, a_t)}{P(s_{t+\tau} = s_* \mid s_t)}\right] \quad (1)$$

The effective empowerment can be used to identify states and actions that have high power. To do so, we define the state-conditional empowerment $\mathcal{E}(s, \pi_{LM})$ for the state $s \in \mathcal{S}$ and state-action conditional empowerment $\mathcal{E}(s, a, \pi_{LM})$ defined for a given state-action pair $(s, a) \in \mathcal{S} \times \mathcal{A}$:

$$\mathcal{E}(s, \pi_{LM}) \triangleq \mathbb{E}_{a \sim \pi_{LM}, s_*}\left[\sum_{t=0}^{\infty} \frac{\gamma^t}{1 - \gamma} \log \frac{P(s_{t+\tau} = s_* \mid s_t = s, a)}{P(s_{t+\tau} = s_* \mid s_t = s)}\right] \quad (2)$$

$$\mathcal{E}(s, a, \pi_{LM}) \triangleq \mathbb{E}_{s_*}\left[\sum_{t=0}^{\infty} \frac{\gamma^t}{1 - \gamma} \log \frac{P(s_{t+\tau} = s_* \mid s_t = s, a_t = a)}{P(s_{t+\tau} = s_* \mid s_t = s)}\right] \quad (3)$$

**Connection Between Empowerment and Agent Capability** Prior work (Myers et al., 2025) shows that empowerment can be interpreted as the expected return under the uniform reward goals assumption: when rewards are randomly drawn across all possible states, empowerment lower-bounds the

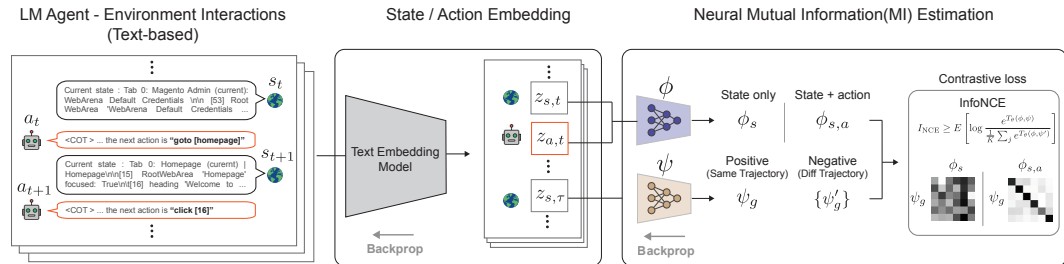

Figure 2: **EELMA Overview.** EELMA quantifies the empowerment of LM-agent from text-based trajectories by mapping textual observations and actions to compact embeddings and estimating variational mutual information using InfoNCE (Le-Khac et al., 2020).

mean discounted reward ($\bar{r} = \mathbb{E}_R[\sum_{t=0}^{\infty} \gamma^t r_t]$, with the MDP discount factor $\gamma$). (see Appendix A.2 for details). Intuitively, this means that an agent with high empowerment is well positioned to succeed across any arbitrary task because it has more action options and pathways into the future. Crucially, these options unfold over multiple turns: empowerment explicitly captures an agent's ability to sustain influence and preserve optionality across a sequence of interactions, rather than a single decision point. This property makes empowerment a principled quantification of agentic capability in multi-turn horizons, where success depends not only on immediate actions but also on maintaining future flexibility. To formalize,

> **Empowerment as proxy for agentic performance.** Empowerment provides a goal-agnostic evaluation metric for LM-agent capability in multi-turn horizons, and empirically serves as a proxy for average goal reward.

This framing allows us to quantify the efficiency of language model agents with a concrete, computable metric. Throughout the paper, we test this claim by comparing mean empowerment with mean discounted reward across a range of agentic tasks, including toy games and realistic multi-turn scenarios such as web browsing.

**The EELMA Algorithm.** Here, our focus is on LM agents navigating in a text-based environment, e.g., natural language, code, web pages, etc. In these environments, states $s \in \mathcal{S}$ and actions $a \in \mathcal{A}$ are both represented using text, which introduces unique challenges. First, unlike the continuous control tasks typical in empowerment literature (Du et al., 2020; Jung et al., 2011), text environments are high-dimensional and combinatorially sparse, making direct calculation of the policy $\pi$ intractable. Second, text is subject to linguistic variability, creating a many-to-one mapping where distinct textual states can share the same underlying symbolic semantics (e.g., paraphrasing 'the agent is at (1,2)' vs. 'located at x=1, y=2'). Since empowerment quantifies an agent's control, it must be estimated at the level of latent semantics rather than surface-level text. To our knowledge, we are the first to quantify empowerment in language spaces.

We propose an algorithm for *Estimating the Empowerment of a Language Model Agent* (**EELMA**). EELMA is an indirect method for quantifying empowerment objective through learning representation (Figure 2). EELMA first maps textual observations and actions into compact embeddings via a language embedding model. Next, we apply variational mutual information estimation e.g., InfoNCE (Le-Khac et al., 2020; Rusak et al., 2025) from this embedding. Motivated by prior work emphasizing compact representations for effective feature extraction (Bharadhwaj et al., 2022; Myers et al., 2025), EELMA enables quantifying the empowerment from text-based trajectories.

For language embedding, given multi-turn trajectories $\{(s_t^i, a_t^i)\}_{t=1}^{T_i}$, where $i = 1, \ldots, N$ enumerates individual trajectories and $t$ indexes steps within each trajectory, we sample tuples consisting of the current state, current action, and future state $((s_t^i, a_t^i, s_*^i))$ from the multi-turn trajectories and map these tuples into embeddings $(z_{s,t}^i, z_{a,t}^i, z_{s_*,t}^i)$ using an embedding model. We use a pretrained embedding model coupled with a fine-tunable MLP (parameterized by $\theta$) that projects to a compact dimension.

For mutual information estimation, we leverage contrastive successor representations method proposed by Myers et al. (2025). We apply two neural encoders: the encoder $\phi$, which encodes the

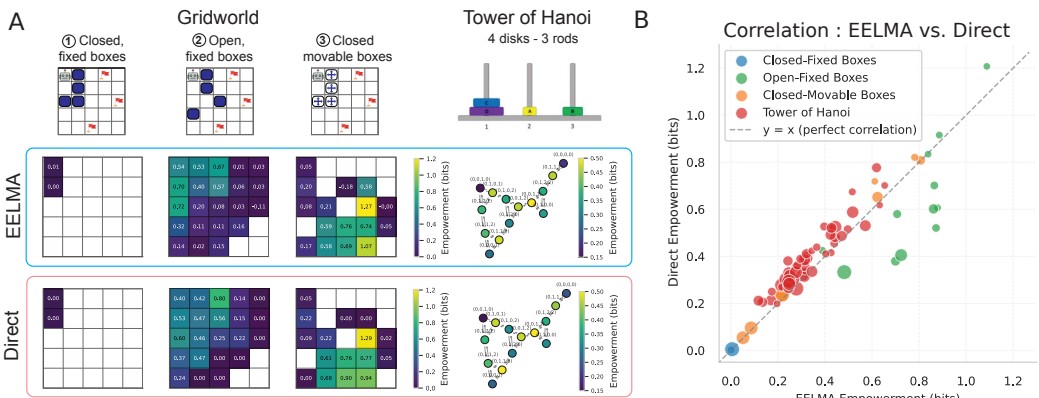

Figure 3: **EELMA accurately estimates the effective empowerment.** We validated the EELMA algorithm in three Gridworld scenarios and the Tower Of Hanoi(ToH). (A) State-conditional empowerment estimated by EELMA closely aligns with direct estimation. Heatmaps represent empowerment averaged across agent positions in the Gridworld. The graphs display empowerment for configuration (merged by permutation symmetry) in the ToH. (B) The correlation plot shows strong alignment between effective empowerment estimates from EELMA and direct estimation.

current state $(z_{s,t})$ and state-action pair $(z_{s,t}, a_{s,t})$, and the encoder $\psi$, which encodes future states $(z_{*,t})$. Using these encoded representations, we compute the InfoNCE loss as follows:

$$\text{State-only } I_{\text{NCE}} \geq \mathbb{E}\left[\log \frac{e^{\phi(z_{s,t}^i)^\top \psi(z_{s,*}^i)}}{\frac{1}{K}\sum_j e^{\phi(z_{s,t}^i)^\top \psi(z_{s,*}^j)}}\right], \quad \text{State-action } I_{\text{NCE}} \geq \mathbb{E}\left[\log \frac{e^{\phi(z_{s,t}^i, z_{a,t}^i)^\top \psi(z_{s,*}^i)}}{\frac{1}{K}\sum_j e^{\phi(z_{s,t}^i, z_{a,t}^i)^\top \psi(z_{s,*}^j)}}\right]. \quad (4)$$

Note that, in the above, negative samples are the target states from the different trajectories. We jointly maximize these two NCE objectives with respect to both encoders $\phi$ and $\psi$, as well as the embedding projection $\theta$. The detailed procedure for estimator training is described in Appendix 1.

To estimate empowerment, we utilize learned representations obtained from embedding model($\theta$) and encoders($\phi, \psi$). Following the work by Myers et al. (2025), learned successor representation is simply converted to mutual information at convergence:

$$\phi(z_{s,t}, z_{a,t})^\top \psi(z_{s,*}) = \log P(s_{t+K} = s_* \mid s_t, a_t) - \log P(s_{t+K} = s_*) - \log C_1 \quad (5)$$

$$\phi'(z_{s,t})^\top \psi(z_{s,*}) = \log P(s_{t+K} = s_* \mid s_t) - \log P(s_{t+K} = s_*) - \log C_2 \quad (6)$$

Thus, our effective empowerment is estimated by averaging the subtract of two dot products:

$$\mathcal{E}(\pi_{LM}) = \mathbb{E}_{i,t,s^*}[\phi(z_{s,t}^i, z_{a,t}^i)^\top \psi(z_{s,*}^i) - \phi(z_{s,t}^i)^\top \psi(z_{s,*}^i)] \quad (7)$$

## 4 RESULTS: EFFECTIVE EMPOWERMENT IN LANGUAGE GAMES

**EELMA in text-based games.** In this section, we validate the EELMA algorithm by answering the research question: Does EELMA accurately model effective empowerment? We answer this question using two highly controlled environemnts: a spatial navigation Gridworld and Tower of Hanoi, a test of reasoning, both implemented with a natural language interface. The tractable state-action space of these environments allows for direct estimation of empowerment via conditional probabilities (refer to Appendix A.1) and comparison with EELMA.

Gridworld, contains three scenarios: (1) an agent enclosed by immovable boxes, (2) an agent with an open route among immovable boxes, and (3) an agent enclosed by boxes that can be moved around. In each scenario, the agent is initialized at the top-left corner of a 5-by-5 grid, and a goal state is a randomly sampled location from the unoccupied squares in the grid. The LM-agent was prompted to reach the goal state. In Tower of Hanoi (ToH), the LM-agent rearranges four different-sized disks across three rods, while following the rule that a larger disk cannot be placed on top of a smaller one, until a goal configuration of disks is reached. Initial and goal states were randomly sampled from

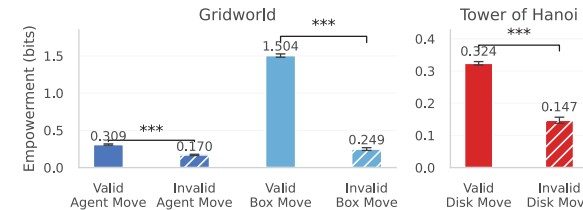

Figure 4: **EELMA identifies influential actions.** State–action conditional empowerment for valid (leading to novel states according to the game rules) and invalid actions in GridWorld (left) and ToH (right). Valid actions, which produce meaningful state transitions (e.g., moving to an empty grid in GridWorld, or placing a smaller disk onto a larger one in ToH), exhibit significantly higher empowerment than invalid actions (e.g., moving into a box, or placing a larger disk onto a smaller disk in ToH). The difference between valid and invalid actions is statistically significant (*** $p < 0.001$, t-test).

the 81 possible configurations. Detailed descriptions of the games are provided in Appendices E, F. A total of 800 trajectories were generated using LM-agents with gpt-4o-mini for Gridworld and claude-3.5-sonnet for ToH.

Across all scenarios, effective empowerment estimates produced by EELMA converge to the ground truth values shown in Figure 9. In Figure 3, we conducted detailed comparisons of state-conditional empowerment between EELMA and direct estimation upon convergence. Empowerment estimated by EELMA visualized by agent location in Gridworld and per symmetrical configuration in ToH, closely matches the direct estimation. Panel B demonstrates strong state-level correlations between EELMA and direct estimation, highlighting the precision of EELMA within these games.

Figure 3 demonstrates how effective empowerment quantifies the *optionality* an agent has within an environment. For example, in scenario 1, the agent has no option beyond bouncing between the two enclosed squares, resulting in a very low empowerment. In contrast, scenario 2 permits the agent to navigate through available spaces, increasing empowerment. Scenario 3 exhibits even higher empowerment, as the agent gains additional options through box-moving actions. Similarly, in the ToH, states with dispersed disks exhibit greater effective empowerment than states where disks are stacked on a single rod, as they allow more possible disk moves. Finally, Figure 4 shows that effective empowerment distinguishes *influential actions* that bring the agent to a novel state from those that do not.

**Robustness and accuracy of EELMA.** EELMA provides reliable empowerment estimates even in regimes where baselines fail, for example when direct estimation collapses under natural-language variability, e.g., when the same state is described as "agent is located at x=2,y=1" versus "agent stands at x,y=2,1."" To test this, we constructed paraphrased variants of GridWorld and Tower of Hanoi using LLM-assisted rephrasings, thereby increasing "language uncertainty" ($H$(observation | latent state)) (Appendix I). Under observations with natural–language variability, *direct estimation* exhibits substantially larger *state* errors in state-conditional effective empowerment estimation in both GridWorld and Tower of Hanoi(Table 1). By contrast, *EELMA (NL)* remains close to its fixed-format baseline for *state* empowerment, indicating robust accuracy under linguistic variability. Together, these results demonstrate that EELMA delivers the accurate estimation with robustness to linguistic variability, enabling it to work effectively for LM agents in language-based environments.

Table 1: **EELMA is robust to natural-language variability**. RMSE (lower is better) of State-conditional predicted effective empowerment compared to DE, reported in *bits*, under structured vs. NL observations for two domains.

|  | State RMSE | |
|---|---|---|
| Method | GridWorld | Tower of Hanoi |
| EELMA (fixed format) | 0.056 | 0.158 |
| **DE (NL observation)** | 0.302 | 0.438 |
| **EELMA (NL observation)** | 0.048 | 0.127 |

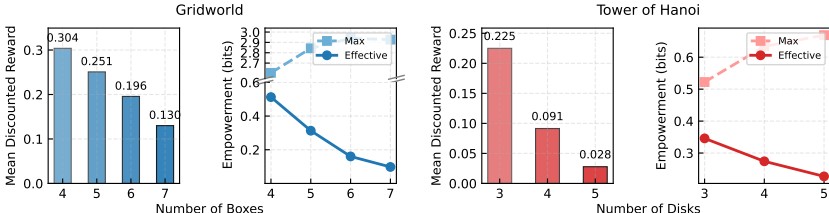

Figure 5: **Environmental complexity affects effective empowerment.** We vary the number of boxes from 4 to 7 in a 4-by-4 Gridworld (left), and the number of disks from 3 to 5 in the ToH of 3 rods (right). The effective empowerment of the LM-agent progressively decreases in environments compared to max empowerment (e.g., theoretical bound that optimal policy can exert influence) in higher complexity, correlating closely with reduced average rewards.

**Effective empowerment is lower when agents struggle in more complex environment.** Figure 5 shows how environmental complexity alters effective agent empowerment. LM-agents have lower average reward in increasingly complex environments such as the presence of more movable boxes in Gridworld or additional disks in the ToH. We compared the effective empowerment to the maximum theoretical value (channel capacity) as calculated using the Blahut–Arimoto algorithm Arimoto (1972); Fasoulakis et al. (2025). For details of the calculations, refer to Appendix B.1.

Our results capture how current LM-agents suffer when increasing the obstacles or dimensions of the game, even if the underlying rules of the game remain unchanged. This finding aligns with previous observations that LM-agents struggle to solve spatial tasks at larger scales (Lin et al., 2025; Bober-Irizar, 2025). Intuitively, human players who rely on an understanding of the game rule would be less affected by scales and maintain their effective empowerment. This contrasts with our observation for LM-agents, highlighting a challenge in preserving empowerment in task at scale.

**Effective empowerment tracks goal-averaged performance over variations of LM-agents.** We next investigate the effective empowerment of LM-agents with various ablations. We specifically study how Chain-of-Thought (CoT) prompting, memory context length, and base LLM, influence effective empowerment and performance. For CoT ablation, we removed the instructions in the prompt to use CoT prior to generating actions. To study the influence of memory, we provide agents with responses at the previous 1, 2, or 3 steps. We also varied base LLM, testing both closed-source models (GPT and Claude models) and open-weight models (Gemma, Qwen, and Llama 3) of varying parameter sizes. Detailed information of ablations is provided in Appendix H.

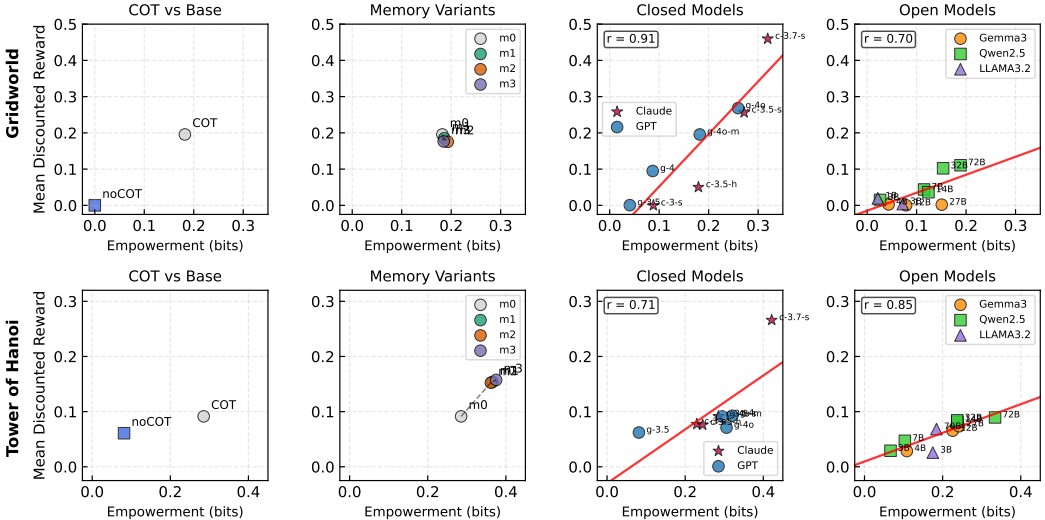

Figure 6: **Empowerment and performance across variations of LM-agents.** We evaluated how Chain-of-Thought (CoT) prompting (first column), memory context length (second column), and the choice between publicly available and closed base models (third and fourth columns) affect effective empowerment and mean discounted reward. Gridworld results are presented in the top row, and ToH results in the bottom row.

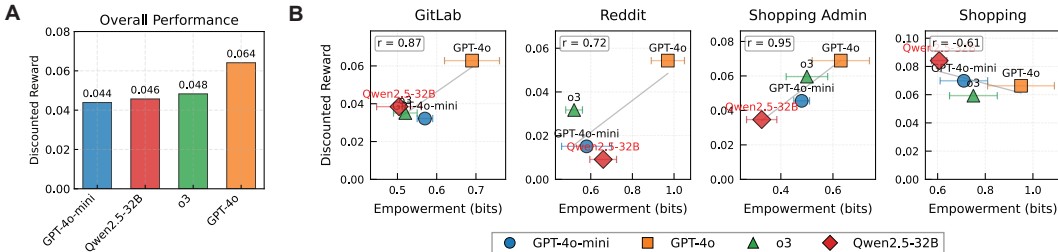

Figure 7: **EELMA in WebArena, a realistic web browsing environment.** We applied EELMA to across four domains of the WebArena benchmark using GPT-4o-mini, GPT-4o, o3, and Qwen2.5-32B. (A) Overall performance across the four domains was quantified using mean discounted reward. (B) Domain-wise empowerment scores computed by EELMA effectively shows strong correlation with discounted rewards. Error bars indicate standard deviation across three different EELMA training seeds.

We collected 1600 trajectories for a 4-by-4 Gridworld environment with 6 movable boxes and 800 trajectories for a ToH environment with 4 disks and 3 rods, each having randomized initial and goal states. Using these trajectories, we estimate effective empowerment with EELMA and plotted it against the mean discounted reward. Figure 6 shows that effective empowerment showed strong correlations with mean discounted reward across different ablations and conditions. The results support our *main claim* that effective empowerment can approximate agentic performance.

Figure 6 shows an impact of different ablations on effective empowerment and performance. Agents exhibit significantly reduced empowerment without CoT reasoning. Disabling CoT drastically reduces empowerment, with a 99% decrease in Gridworld (from 0.19 to 0.01 bits) and a 65% decrease in ToH (from 0.29 to 0.09 bits). Increasing memory context length increases empowerment and performance. We observed that extending the agent's memory from 0 to 3 previous steps (m0 to m3) progressively increased empowerment, particularly evident in the ToH environment, where empowerment rose from approximately 0.3 to 0.4 bits with additional memory. Closed-weight LLMs generally exhibit higher empowerment than open-weight LLMs and effective empowerment scales positively with model size and release version. Among open-source models, Qwen2.5 exhibited clear parameter-scaling behavior, whereas Gemma-3 and LLaMa-2 did not. Within closed-source models, higher-version models (e.g Claude-3 Sonnet vs. Haiku; GPT-4o vs. GPT-4o-mini) consistently demonstrated superior empowerment and performance.

**Implementation Guidelines:** We investigate how base encoder choice, degree of fine-tuning, and computational cost trade off in practice; First, we find that LoRA adaptation of the encoder offers the best accuracy–stability–cost trade-off, improving RMSE over a frozen encoder while avoiding the training collapse observed for partial or full fine-tuning and adding only a few MB of parameters. Second, base encoder choice matters: larger models such as E5-Base-v2 generally improve performance, but compact architectures like MiniLM-L6-v2 can perform best, suggesting that sentence-level embedding quality is more important than parameter count. Detailed results are reported in the Appendix K, L.

# 5 RESULTS: EFFECTIVE EMPOWERMENT IN WEB ENVIRONMENT

In this section, we use EELMA to study empowerment in WebArena (Zhou et al., 2023), a realistic web-browsing environment designed to support open-world interactions. Again, our goal is to assess whether effective empowerment serves as a goal-agnostic proxy for agent performance.

EELMA was trained to quantify the effective empowerment of LM-agents across four domains (GitLab, Reddit, Shopping Admin, and Shopping) of the WebArena benchmark using three closed-source models (GPT-4o-mini, GPT-4o, and o3) and Qwen2.5-32b-it. Agents are tasked with realistic goals (e.g., identifying the price range of a *Canon Photo Printer* in an online shopping mall) and navigate based on observations drawn from the HTML DOM tree. In addition to the original tasks provided by Zhou et al. (2023), we augment the task set with randomly generated goals created by large language models to obtain more diverse trajectories. These augmented trajectories are used

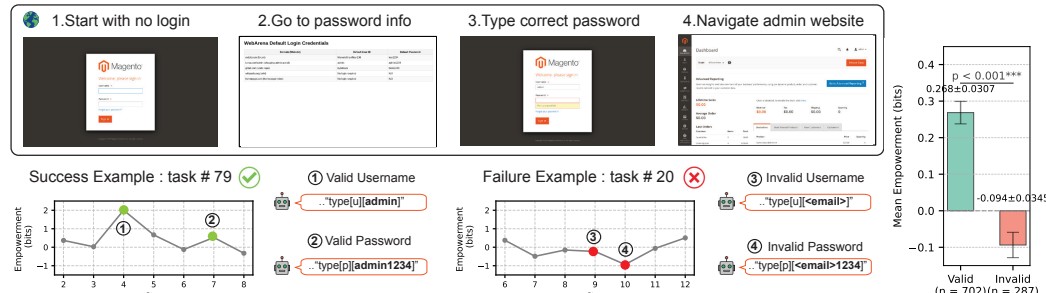

Figure 8: **EELMA Captures Valid Authentication Actions.** We analyzed state-action empowerment estimates for authentication behaviors(username/password typing). Typing valid usernames and passwords resulted in high empowerment, whereas invalid actions did not (Right panel).

for EELMA estimation but are not counted as part of the reward. A detailed description of the experimental setup is in Appendix G.

Figure 7A shows the overall performance of three models across domains. We find that GPT-4o has the highest discounted reward compared to o3, GPT-4o-mini, and Qwen. When just looking at task success, we find that o3 has a comparable success rate to GPT-4o but it takes a larger number of steps to reach the goal states (Figure C.1) leading to lower discounted reward. Consistent with these observations, effective empowerment estimated by EELMA across all four domains shows that GPT-4o has the highest influence on the environment. Figure 7B shows a strong correlation between mean discounted reward and estimated empowerment in the GitLab, Reddit, and Shopping Admin domains (Rs=0.83-0.94). Together, these results show that effective empowerment serves as an indicator of agentic capability in a realistic open world environment.

In contrast, in the Shopping tasks there was a flat relationships between discounted reward and effective empowerment (Figure 7B). In the Shopping task (e.g., identifying the price range of a *Canon Photo Printer* in an online shopping mall), the agent must not only navigate through the environment efficiently but also perform reasoning about numerical prices. Such reasoning capabilities might represent a bottleneck that limits performance, regardless of empowerment. Consistent with this possibility, the estimated empowerment values for the Shopping domain are already relatively high suggesting that empowerment over the environment is not a limiting factor for performance.

Interestingly, Qwen performs poorly on Reddit tasks yet maintains quite high empowerment comparable to GPT-4o-mini and o3 (Figure 7), showing a non-negligible offset from the linear fitting line. We found this is due to 'jailbreaking' behavior: 40% of Qwen's trajectories navigate to external websites (e.g., the real "www.reddit.com") rather than the WebArena sandbox server, shown in Figure 11b in Appendix. Consequently, the model navigates to diverse but task-irrelevant external websites; this artificially inflates state diversity and empowerment estimates, even though the agent fails the task objectives.

**Case Study: Power-seeking and Authentication** Finally, we demonstrate how effective empowerment can detect pivotal actions or situations where an agent is accessing more resources than intended (Turner et al., 2021; Turner & Tadepalli, 2022). We created a "modified shopping admin" environment, where authentication is not automatically provided for the agent. To successfully complete the shopping admin tasks, the agent must first navigate the website, locate the username and password information on a hidden page, and manually enter these credentials to log in to the shopping admin main panel (Figure 8). In addition to authentication, the LM agent also performed the original WebArena tasks (n = 182) in the shopping admin domain.

Intuitively, successful authentication should be a key moment where effective empowerment should increase. Once authenticated, the agent has access to (and control over) much more of the environment. Thus, we hypothesize that successful authentication-related actions would result in higher effective empowerment, whereas invalid authentication attempts would have lower in effective empowerment. There are no rewards associated with either of these steps in WebArena.

We observe that GPT-4o (with 1-step memory) successfully figures out how to authenticate itself 137 times out of 182 trajectories. GPT-4o without memory and GPT-4o-mini both fail to authenticate (Table 4). Figure 8 illustrates representative trajectories for successful and unsuccessful account-

authentication attempts. Effective empowerment sharply increases when the agent enters a valid username and password, whereas it remains low during invalid attempts. Across all 182 trajectories, the mean empowerment scores for typing actions for valid authentication were 0.268 bits, which are higher than the scores of $-0.094$ bits for typing actions for invalid authentication, with a significance of $p < 0.001$ (Figure 10). Interestingly, username typing shows a smaller difference in empowerment between valid and invalid (0.204 bits) entries with no statistical significance ($p = 0.32$), compared to password typing where valid entries (0.154 bits) significantly exceed invalid attempts ($-0.112$ bits, $p < 0.001$). Note that the negative empowerment values arise from the InfoNCE-based approximation, as detailed in Appendix G.3.. This pattern may be explained by the sequential nature of authentication: the agent types the username first, and even a valid username paired with an incorrect password results in no effective gain in future state accessibility, making the password entry more critical. Together, these results suggest that effective empowerment can be leveraged for detecting and monitoring highly empowered behaviors (e.g., taking control over system-administration privileges or gaining access to a restricted domain) without needing to explicitly enumerate these behaviors in advance.

## 6 DISCUSSION

Our study introduces EELMA, a novel algorithm that provides a goal-agnostic evaluation of LM-agent capability using an information-theoretic approach based on empowerment. We show that these EELMA estimates consistently correlate with goal-averaged performance across diverse experimental setups and agent configurations. Thus, EELMA gives a goal-agnostic measure of agent capability. Unlike conventional evaluation benchmarks, our method requires no explicit goal annotations. Future research could extend this method to multi-agent scenarios. For additional details on multimodal extensions and experiments on power-seeking behavior, see Appendix M.

**Limitations** The scope of our work is limited to the empowerment metric, which quantifies an agent's control over future states based on the number of options (alternative futures) the agent can meaningfully access or influence. However, having more options does not always translate directly into greater power. For instance, having one strong job offer can be more advantageous than multiple poor offers during salary negotiations. Additionally, empowerment does not capture other forms of power, such as indirect power, i.e., influence over other agents' beliefs, decisions, and actions.

The "curse of dimensionality" is a key challenge for scaling empowerment to more complex, longer-horizon tasks. In a dense MDP, the number of possible trajectories grows exponentially with the horizon $T$ (roughly $O(|A|^T)$), and therefore the number of rollouts required for precise empowerment estimation also scales exponentially. However, we argue that this worst-case complexity rarely manifests in practice: in real-world tasks the effective branching factor is much smaller because only a sparse subset of actions leads to meaningful state transitions or non-zero rewards. EELMA exploits this sparsity, enabling efficient estimation even in longer-horizon settings without exploring the full exponential trajectory space.

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

# A THEORETICAL FOUNDATION

We provide the theoretical foundations for effective empowerment as supplementary material to Section 3.

## A.1 DIRECT ESTIMATION OF EFFECTIVE EMPOWERMENT

Figure 3 compares EELMA with direct empowerment estimation computed explicitly from a forward dynamics model. The procedure is detailed below:

Given a dataset of $N$ trajectories $\{(s_t^{(i)}, a_t^{(i)})\}_{t=0}^{T_i}$, $i = 1, \ldots, N$, we estimate empowerment at state $s$ by constructing an empirical forward dynamics model $\hat{p}(s_* \mid s, a)$.

Define the count of observed transitions from state $s$ to successor state $s_*$ via action $a$ across all trajectories as:

$$N(s, a, s_*) = \sum_{i=1}^{N} \sum_{t=0}^{T_i - 1} \mathbb{I}(s_t^{(i)} = s,\ a_t^{(i)} = a,\ s_*^{(i)} = s_*)$$

Next, define the total occurrences of action $a$ taken in state $s$ as:

$$N(s, a) = \sum_{s_*} N(s, a, s_*)$$

Then, the forward dynamics probabilities are estimated via Maximum Likelihood Estimates(MLE):

$$\hat{p}(s_* \mid s, a) = \frac{N(s, a, s_*)}{N(s, a)}$$

With $\hat{p}(s_* \mid s, a)$ computed, empowerment is defined as the mutual information between actions and successor states:

$$\hat{\mathcal{E}}(s) = I(A; S_* \mid s)$$

where $A$ denotes the action, and $S_*$ is the resulting successor state conditioned on state $s$.

## A.2 EMPOWERMENT AS PROXY FOR POWER

We take advantage of the theoretical results of relation between effective empowerment and average-goal performance of Myers et al. Myers et al. (2025) with slight adjustment to our case where only the LLM policy $\phi_{\text{LLM}}$ exists. The three assumptions are required to provide the connection between empowerment and goal :

**Assumption : Skill Coverage** The rewards $R \sim \mathcal{R}$ are uniformly distributed over the scaled $|\mathcal{S}|$-simplex $\Delta^{|\mathcal{S}|}$, such that:

$$\left(R + \frac{1}{|\mathcal{S}|}\right) \frac{1}{1 - \gamma} \sim \text{Unif}\left(\Delta^{|\mathcal{S}|}\right) = \text{Dirichlet}(1, 1, \ldots, 1).$$

This assumption implies the reward function is uniform over the states in the environment, effectively diverse skills are related to goal-average performance.

**Assumption : Ergodicity** For some human policy $\pi_H$ and robot policy $\pi_R$, it holds that:

$$\mathbb{P}^{\pi_{LLM}}(s_* = s \mid s_0) > 0 \quad \text{for all } s \in \mathcal{S},\ \gamma \in (0, 1).$$

This guarantees that under the joint policies $\pi_H$ and $\pi_R$, every state $s$ in the state space $\mathcal{S}$ is reachable from the initial state $s_0$ with positive probability, ensuring sufficient exploration of the state space.

**Assumption : Boltzman rationality of agent** The LLM agent is assumed to be Boltzmann-rational with respect to the robot's policy. Specifically, the probability of the LLM agent selecting a

sequence of actions $a_t, \ldots, a_{t+\tau}$ given the current state $\bar{s}_t$ and reward function $R$ is proportional to the exponentiated expected cumulative reward:

$$\mathbb{P}(a_t, \ldots, a_{t+\tau} \mid \bar{s}_t, R) \propto \exp\left(\beta \cdot \mathbb{E}\left[\sum_{k=0}^{\tau} \gamma^k R(s_{t+k}, a_{t+k})\right]\right),$$

where $\beta > 0$ is the rationality coefficient, $\gamma \in (0, 1)$ is the discount factor, and the expectation is taken over state transitions induced by the LLM agent's and robot's policies.

Under these assumptions, we derive the following lemma:

**Lemma 1**  Let $\tau \sim \mathrm{Geom}(1-\gamma)$ and $\tau \geq 0$. Then,

$$\liminf_{\gamma \to 1} I(s_*; a_t, \ldots, a_{t+\tau} \mid s_t) \leq I(R; a_t, \ldots, a_{t+\tau} \mid \bar{s}_t),$$

where $s_\gamma^+$ denotes the future state at time $t$ under discount factor $\gamma$, $a_t, \ldots, a_{t+\tau}$ are the LLM agent's actions from time $t$ to $t+\tau$, $\bar{s}_t$ is the state at time $t$, and $R$ represents the reward function.

**Proof:** We refer to Myers et al. Myers et al. (2025) for a detailed proof; here, we provide a brief sketch. For sufficiently large $\gamma$, the future state $s_\gamma^+$ approaches the stationary distribution induced by the joint policies $(\pi_{\mathrm{LLM}}, \pi_R)$, irrespective of the current state $s_t$ and actions $a_t, \ldots, a_{t+\tau}$, as guaranteed by Assumption A.2. Thus, we have:

$$\liminf_{\gamma \to 1} I(s_*; a_t, \ldots, a_{t+\tau} \mid s_t) \ \leq \ I\left(\lim_{\gamma \to 1} s_*; a_t, \ldots, a_{t+\tau} \mid s_t\right).$$

Next, the Boltzmann rationality assumption (Assumption A.2) guarantees that the LLM agent's policy $\pi_{\mathrm{LLM}}$ induces the following Markov chain structure:

$$\hat{a}_t \ \longrightarrow \ R \ \longrightarrow \ \lim_{\gamma \to 1} s_*.$$

Applying the data processing inequality, we obtain:

$$I\left(\lim_{\gamma \to 1} s_*; a_t, \ldots, a_{t+\tau} \mid s_t\right) \ \leq \ I(R; a_t, \ldots, a_{t+\tau} \mid s_t),$$

which completes the proof.

Now to correlate the goal-averaged rweard, Given the LLM agent's policy $\pi_{\mathrm{LLM}}$, reward function $R$, and discount factor $\gamma \in (0, 1)$, the soft Q-function for a state-action trajectory $(s_t, a_t, \ldots, a_{t+\tau})$ is defined as:

$$Q_{R,\gamma}^{\pi_{\mathrm{LLM}}}(s_t, a_t, \ldots, a_{t+\tau}) \triangleq \mathbb{E}_{\pi_{\mathrm{LLM}}}\left[\sum_{k=0}^{\tau} \gamma^k \left(R(s_{t+k}, a_{t+k}) - \frac{1}{\beta}\log \pi_{\mathrm{LLM}}(a_{t+k} \mid s_{t+k})\right) \,\middle|\, s_t, a_t, \ldots, a_{t+\tau}\right],$$

where the expectation is taken over future state-action transitions under the LLM agent's policy $\pi_{\mathrm{LLM}}$, and $\beta > 0$ is the rationality coefficient.

**Lemma 2** For any time $t$ and horizon $\tau \geq 0$, the following inequality holds:

$$I(R; a_t, \ldots, a_{t+\tau} \mid s_t) \leq \lim_{\gamma \to 1}\left(\frac{\beta}{e}\mathbb{E}\left[Q_{R,\gamma}^{\pi_{\mathrm{LLM}}}(s_t, a_t, \ldots, a_{t+\tau})\right]\right)^2,$$

where $Q_{R,\gamma}^{\pi_{\mathrm{LLM}}}(s_t, a_t, \ldots, a_{t+\tau})$ denotes the soft Q-value under reward function $R$, discount factor $\gamma$, LLM agent policy $\pi_{\mathrm{LLM}}$, and robot policy $\pi_R$; $\beta$ is the rationality coefficient, and $e$ is Euler's number.

**Proof:** We refer to Lemma B3 in Myers et al. Myers et al. (2025) for a detailed proof.

**Theorem** Based on Lemma 1 and Lemma 2, we deduce the following lower-bound relationship for empowerment at sufficiently large $\gamma$:

$$\mathcal{E}_\gamma(\pi_{\mathrm{LLM}})^{1/2} \ \leq \ \frac{\beta}{e}\mathcal{J}_R^\gamma(\pi_{\mathrm{LLM}}),$$

where

$$\mathcal{J}_R^\gamma(\pi_{\text{LLM}}) = \mathbb{E}\left[V_{R,\gamma}(\pi_{LLM})\right] = \mathbb{E}\left[\sum_{t=0}^\infty \gamma^t \left(R(s_t, a_t) - \frac{1}{\beta}\log \pi_{\text{LLM}}(a_t \mid s_t)\right)\right],$$

where $\mathcal{E}_\gamma(\pi_{\text{LLM}})$ represents the empowerment objective, and $\mathcal{J}_R^\gamma(\pi_{\text{LLM}})$ means expected discounted cumulative reward under policy $\pi_{\text{LLM}}$. This indicates that goal-averaged discounted reward can be lower bounded by the effective empowerment, establishing a quantifiable connection between empowerment and reward-driven objectives.

### A.3 EMPOWERMENT IN PARTIALLY OBSERVABLE MARKOV DECISION PROCESS(PODMP)

Although our work assumes a fully observable Markov Decision Process (MDP) as the main framework, the empowerment objective can readily be extended to partially observable Markov decision processes (POMDPs). In prior works, empowerment originally quantifies an agent's control over future sensor observations through its actions. Formally, the modified empowerment definition can be expressed as follows:

$$\mathcal{E} = \mathbb{E}[I(o_*, a_i \mid o_i)]$$

where $o_i$ denotes the current observation, $a_i$ the current action, and $o_*$ the future observation.

## B SUPPLEMENTARY : LANGUAGE GAMES

Here, we provide supplementary information to support the WebArena results in Section 4.

### B.1 MAXIMUM EMPOWERMENT CALCULATION

The maximum empowerment for a given state is calculated using the Blahut-Arimoto algorithm Fasoulakis et al. (2025), which iteratively optimizes mutual information (MI) between actions and the resulting future states. Specifically, starting from an initial Tower of Hanoi configuration, the algorithm samples possible future states by repeatedly performing valid or optionally including invalid actions according to geometric discounting with factor $\gamma = 0.9$. At each iteration, the conditional probabilities of future states given actions, $p(s|a)$, are empirically estimated from the trajectories sampled. The Blahut-Arimoto algorithm then alternates between updating the action distribution $p(a)$ to maximize MI and recalculating state distributions until convergence, indicated by changes in MI falling below a threshold of $\delta = 10^{-6} bit$.

### B.2 EELMA TRAINING

We trained the EELMA model for approximately 10,000 optimization steps, observing stable convergence within this training regime as shown in Figure 9.

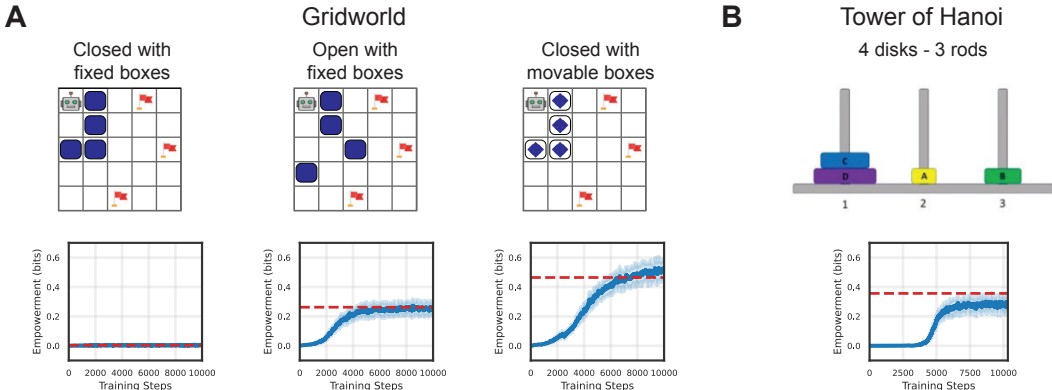

Figure 9: **Convergence of Empowerment Estimates in Gridworld and Tower of Hanoi Tasks.** Empowerment estimates similarly reach convergence by 10,000 training steps for (A) three Gridworld scenarios and (B) Tower of Hanoi task. Red dashed lines indicate asymptotic empowerment levels by direct calculation. Shaded areas represent standard deviations across runs.

## C SUPPLEMENTARY : WEBARENA

Here, we provide supplementary information for Webarena results in Section 5.

### C.1 DOMAIN SPECIFIC SUCCESS RATES

Table 2 presents the raw experimental outcomes from the WebArena experiment, including success counts, success rates, and discounted rewards, broken down by domain and model. Table 3 reports the empowerment values estimated by EELMA using three different random training seeds. The means and standard deviations in Table 3 correspond to those shown in Figure 7.

| Model | Domain | Count | Success Count | Mean Trajectory Length (Success Only) | Success Rate | Discounted Reward |
|---|---|---|---|---|---|---|
| gpt-4o-mini | shopping | 187 | 29 | 17.83 | 0.1551 | 0.06983 |
| gpt-4o | shopping | 187 | 24 | 18.44 | 0.1283 | 0.06628 |
| o3 | shopping | 187 | 28 | 21.74 | 0.1497 | 0.05930 |
| gpt-4o-mini | shopping_admin | 182 | 17 | 17.23 | 0.0934 | 0.04555 |
| gpt-4o | shopping_admin | 182 | 27 | 15.21 | 0.1484 | 0.06889 |
| o3 | shopping_admin | 182 | 31 | 20.85 | 0.1703 | 0.05961 |
| gpt-4o-mini | gitlab | 180 | 20 | 19.33 | 0.1111 | 0.03217 |
| gpt-4o | gitlab | 180 | 26 | 18.32 | 0.1444 | 0.06281 |
| o3 | gitlab | 181 | 22 | 15.35 | 0.1215 | 0.03511 |
| gpt-4o-mini | reddit | 106 | 5 | 21.23 | 0.0472 | 0.01510 |
| gpt-4o | reddit | 106 | 15 | 13.61 | 0.1415 | 0.05434 |
| o3 | reddit | 105 | 18 | 19.69 | 0.1714 | 0.03176 |

Table 2: **Domain-specific WebArena Raw Data**.

| Model | Domain | Emp1 | Emp2 | Emp3 | Mean Empowerment (bits) | Std |
|---|---|---|---|---|---|---|
| gpt-4o-mini | gitlab | 0.423 | 0.423 | 0.406 | 0.4173 | 0.0098 |
| gpt-4o-mini | reddit | 0.472 | 0.426 | 0.366 | 0.4213 | 0.0532 |
| gpt-4o-mini | shopping | 0.544 | 0.483 | 0.461 | 0.4960 | 0.0430 |
| gpt-4o-mini | shopping_admin | 0.354 | 0.342 | 0.371 | 0.3557 | 0.0146 |
| gpt-4o | gitlab | 0.556 | 0.489 | 0.480 | 0.5083 | 0.0415 |
| gpt-4o | reddit | 0.760 | 0.715 | 0.656 | 0.7103 | 0.0522 |
| gpt-4o | shopping | 0.712 | 0.680 | 0.672 | 0.6880 | 0.0212 |
| gpt-4o | shopping_admin | 0.462 | 0.458 | 0.446 | 0.4553 | 0.0083 |
| o3 | gitlab | 0.396 | 0.387 | 0.367 | 0.3833 | 0.0148 |
| o3 | reddit | 0.399 | 0.394 | 0.367 | 0.3867 | 0.0172 |
| o3 | shopping | 0.600 | 0.578 | 0.481 | 0.5530 | 0.0633 |
| o3 | shopping_admin | 0.421 | 0.336 | 0.328 | 0.3617 | 0.0515 |

Table 3: **Empowerment estimates statistics** : mean empowerment, and standard deviation across WebArena domains for different models.

## C.2 CASE STUDY - AUTHENTIFICATION ABLATIONS

Table 4 shows the by gpt-4o-mini with 1- memory, gpt-4o with and without 1-memory. We observe that gpt-4o without memory completely fails. Furthermore, gpt-4o-mini with 1-memroy completely fails too. Observation implies that combinations of certain capabiltiies (memory and reasoning abltiy by model scale) is required for performing such authentification task.

| Model | Domain | Count | Login Success Count | Trajectory Length (Success Only) | Success Rate (%) |
|---|---|---|---|---|---|
| gpt-4o with no memory | modified shopping admin | 20 | 0 | N.A. | 0 |
| gpt-4o-mini | modified shopping admin | 182 | 0 | N.A. | 0 |
| gpt-4o with 1-memory | modified shopping admin | 182 | **137** | 11.84 | **75.27** |

Table 4: **Authentication Success Rates in Modified Shopping Admin Environment.** GPT-4o with 1-memory achieves substantial authentication success (**75.27%**) with shorter average trajectory lengths, while GPT-4o with no memory and GPT-4o-mini fail entirely (**0%**).

Figure 10 shows the empowerment results for valid action typing in the modified shopping WebArena environment, using GPT-4o with 1-memory.

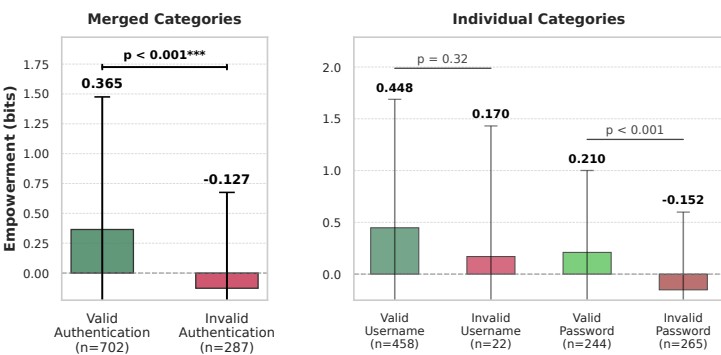

Figure 10: **Empowerment (bits) by authentication action category.** Left panel shows merged categories comparing all valid authentication actions (n=702) versus invalid attempts (n=287), with valid actions showing significantly higher mean empowerment (0.365 bits vs -0.127 bits, $p < 0.001$). Right panel breaks down empowerment scores by specific action types: valid username entries (0.448 bits, n=458) show higher empowerment than invalid username entries (0.170 bits, n=22, $p = 0.32$, non-significant due to small sample size), while valid password entries (0.210 bits, n=244) demonstrate significantly higher empowerment than invalid password attempts (-0.152 bits, n=265, $p < 0.001$). Error bars represent standard deviations. Statistical significance determined by two-tailed Welch's t-test. These results demonstrate that effective empowerment can detect pivotal power-seeking behaviors without explicit reward signals.

# D METHODOLOGY : EELMA

Here, we provide a detailed description of the EELMA setup, including its algorithm, network architecture, loss function, training hyperparameters, and computational resources. The EELMA code is provided anonymously at `https://anonymous.4open.science/r/EELMA-E227`.

## D.1 EELMA TRAINING ALGORITHM

The below Algorithm 1 describes the EELMA training algorithm.

### NETWORK ARCHITECTURE DETAILS

**Base Embedding Model:** We use pretrained language embedding models as the foundation for encoding textual observations and actions. Specifically, for language games (Gridworld and Tower

---

**Algorithm 1** EELMA Training Procedure

---

**Require:** Pretrained LM embedding $\text{Emb}_{init}$, trajectories $\{(s_t^i, a_t^i, s_*^i)\}_{i=1,t=1}^{N,T_i}$, embedding dimension $d$, batch size $K$

1: Initialize embedding model $\text{Emb}_\theta$ using pretrained $\text{Emb}_{init}$ and a fine-tunable MLP $\theta$.
2: Initialize neural encoders $\phi, \psi$ parameterized by $\theta$.
3: **for** each training iteration **do**
4:     Sample minibatch of tuples $\{(s_t^i, a_t^i, s_*^i)\}_{i=1}^K$ from trajectories.
5:     Compute embeddings:

$$z_{s,t}^i = \text{Emb}_\theta(s_t^i), \quad z_{a,t}^i = \text{Emb}_\theta(a_t^i), \quad z_{s_*,t}^j = \text{Emb}_\theta(s_*^j)$$

6:     Compute encoder representations:

$$\phi(z_{s,t}^i), \quad \phi(z_{s,t}^i, z_{a,t}^i), \quad \psi(z_{s_*,t}^j)$$

7:     Compute joint InfoNCE loss:

$$\mathcal{L} = -\frac{1}{K} \sum_{i=1}^K \left[ \log \frac{e^{\phi(z_{s,t}^i)^\top \psi(z_{s_*,t}^i)}}{\frac{1}{K} \sum_j e^{\phi(z_{s,t}^i)^\top \psi(z_{s_*,t}^j)}} + \log \frac{e^{\phi(z_{s,t}^i, z_{a,t}^i)^\top \psi(z_{s_*,t}^i)}}{\frac{1}{K} \sum_j e^{\phi(z_{s,t}^i, z_{a,t}^i)^\top \psi(z_{s_*,t}^j)}} \right]$$

8:     Update parameters $\theta$ to minimize $\mathcal{L}$.
9: **end for**
10: **return** Trained embedding model $\text{Emb}_\theta$ and encoders $\phi, \psi$.

---

of Hanoi), we employ `intfloat/e5-small-v2` Wang et al. (2024), and for WebArena (which requires longer context length), we use `jinaai/jina-embeddings-v2-small-en` Günther et al. (2023). On top of these embedding models, a single fine-tunable MLP projection (parameterized by $\theta$) to a compact representation dimension $d_{emb} = 32$.

**State and Action Encoders ($\phi$, $\psi$):** On top of these embeddings, we define two simple neural encoders, $\phi$ for state and state-action pairs, and $\psi$ for future states. Each encoder is implemented as a two-layer MLP with hidden dimension $d_{hidden} = 128$ and final representation dimension $d_{repr} = 32$ (32 x 128 x 128 x 32).

**Successor Representation and Mutual Information Objective:** We combine state and action embeddings by simple addition to obtain the joint representation used in the InfoNCE loss. Given a batch of $N$ samples $(s_i, a_i, s_*)$, we maximize mutual information $I(A; S_* \mid S)$ using the contrastive InfoNCE loss:

$$\mathcal{L}_{\text{InfoNCE}} = -\frac{1}{N} \sum_{i=1}^N \log \frac{\exp(\phi(z_{s,i}, z_{a,i})^\top \psi(z_{s',i})/\tau)}{\sum_{j=1}^N \exp(\phi(z_{s,i}, z_{a,i})^\top \psi(z_{s',j})/\tau)}, \tag{8}$$

where $\tau$ is a temperature hyperparameter controlling the sharpness of the distribution and is updated over training.

**Training Configuration** Training was performed using the Adam optimizer with an initial learning rate of $2 \times 10^{-4}$, decayed linearly throughout the training, and a batch size of $N = 256$. Gradient clipping with a norm threshold of $1.0$ was applied to ensure training stability. The temperature parameter ($\tau$) is initialized at $1.0$ and is adaptively trainable, decreasing over the course of training. Optimization for these components utilized the Adam optimizer with a fixed learning rate of $lr = 10^{-4}$. All EELMA training were conducted on an NVIDIA A100 GPU with 80GB of memory, and convergence typically occurred within approximately 4 hours.

# E  TASK : GRIDWORLD

## E.1  TASK DESCRIPTION

The Gridworld task involves navigating an agent within a structured 5×5 grid environment, aiming to reach a predefined goal position. At each step, the agent can perform exactly one action, which involves either moving itself or moving an adjacent box by exactly one grid cell in any of the four cardinal directions (up, down, left, or right). Moves are classified as either valid or invalid: valid moves successfully relocate the agent or box into an empty adjacent cell within the grid bounds, while invalid moves occur when the target cell is either occupied by another box or lies outside the grid boundaries. Invalid moves result in no changes to the positions of either the agent or any boxes.

- Valid moves: Moving the top disk from one rod onto either an empty rod or onto a rod where the top disk is larger.

- Invalid moves: Attempting to place a larger disk onto a smaller disk, or attempting to move disks that are not positioned at the top of their rod. Invalid moves result in no change to the current disk arrangement.

### E.1.1  TASK CONFIGURATION

The model used is `gpt-4o-mini`, with `tensor_parallel_size=2` and a random seed `seed_num=1600` for reproducibility. All sessions saved both the agent logs and playthroughs for later analysis.

The basic configuration for experiments in Figure 3:

- `grid_size`: 5 x 5
- `num_boxes`: 4
- `block_goal`: False
- `allow_box_moving`: True
- `init_mode_agent`: random
- `init_mode_boxes`: random
- `chain_of_thought`: Enabled (CoT=1)

The basic configuration for experiments in Figure 4,5,6:

- `grid_size`: 4 x 4
- `num_boxes`: 4,5,6,7 (Varying)
- `block_goal`: False
- `allow_box_moving`: True
- `agent_init_position`: random
- `boxes_init_position`: random
- `chain_of_thought`: Enabled (CoT=1)

The model used is `gpt-4o-mini`, with `tensor_parallel_size=2` and a random seed `seed_num=1600` for reproducibility. All sessions saved both the agent logs and playthroughs for later analysis.

### E.1.2  PROMPT TEMPLATES

**System Message Template**

You are an intelligent agent on a {grid_size} x {grid_size} grid (origin at (0,0) in the bottom-left, where the first index represents the horizontal coordinate increasing to the right, and the second index represents the vertical coordinate increasing upward). Your goal is to reach {agent_goal} by navigating the grid and moving boxes when needed.

**1. Movement:** Allowed directions: Left, Up, Right, Down. - Left: decrease the first index. - Up: increase the second index. - Right: increase the first index. - Down: decrease the second index. You cannot move outside the grid or into a cell occupied by a box.

**2. Entities:** - Agent: Your character, occupying a single cell. - Boxes: Movable objects. Boxes can be pushed to adjacent cells. Boxes cannot overlap with each other or with the agent.

**3. Actions:** - Respond in plain text. - For agent movement, use: `"Move <direction>"` (e.g., "Move Left"). - For box movement, use: `"Move the Box <box_id> <direction>"` (e.g., "Move the Box 3 Left"). Note: You can only move a box when it is adjacent to you; otherwise, nothing happens.

**4. Examples:** - Agent Movement: - From (1,0) to (0,0) (left): `"Move Left"` - From (0,0) to (0,1) (up): `"Move Up"` - From (0,0) to (1,0) (right): `"Move Right"` - From (1,1) to (1,0) (down): `"Move Down"` - Box Movement: - Move Box 1 from (2,0) to (1,0) (left): `"Move the Box 1 Left"` - Move Box 2 from (3,1) to (4,1) (right): `"Move the Box 2 Right"` - Invalid Movements: - Moving out of bounds (e.g., "Move Down" from (0,0)) is invalid. - Attempting to move into a cell occupied by a box is invalid. - Attempting to move a box that is not adjacent is invalid.

---

**Observation Prompt Template**

Step {step} Observation: Agent location: {agent_location}, Boxes location: {boxes_location}

---

The agent is instructed to engage in explicit **Chain-of-Thought (CoT)** reasoning before selecting an action. The instruction prompt is:

**Instruction Prompt Template**

Step {step}: Please think through your reasoning step by step (Chain of Thought) and then decide the best action. Select the single best action and provide your response in the following format:
Reasoning: <your detailed reasoning here>
Action: `"Move <direction>"` or `"Move the Box <box_id> <direction>"`

# F    TASK : TOWER OF HANOI

## TASK DESCRIPTION

The Tower of Hanoi task involves rearranging disks across three rods, aiming to transform an initial random disk configuration into a specified goal arrangement. The environment consists of 3 rods labeled $A, B, C$ and 4 disks of varying sizes. Initially, these disks are stacked onto the rods, adhering to the rule that larger disks must always be positioned below smaller disks.

At each step, the agent generates an action by moving exactly one disk from the top of one rod to the top of another rod or onto an empty rod. Moves are classified into valid or invalid according to the following constraints:

- Valid moves: Moving the top disk from one rod onto either an empty rod or onto a rod where the top disk is larger.

- Invalid moves: Attempting to place a larger disk onto a smaller disk, or attempting to move disks that are not positioned at the top of their rod. Invalid moves result in no change to the current disk arrangement.

Both initial and goal configurations are randomly sampled from all permissible arrangements, ensuring diverse task conditions. At each step, the agent receives structured observations explicitly detailing the current and goal configurations.

## TASK CONFIGURATION

The basic configuration for experiments in Figure 3:

- `num_rods`: 3
- `num_disks`: 4
- `init_configuration`: random
- `target_configuration`: random
- `chain_of_thought`: Enabled (CoT=1)

The basic configuration for experiments in Figure 4,5,6:

- `num_rods`: 3
- `num_disks`: 3,4,5 (Varying)
- `init_configuration`: random
- `target_configuration`: random
- `chain_of_thought`: Enabled (CoT=1)

PROMPT TEMPLATES

The agent receives a **system message** that defines the game setup, movement rules, and examples of valid and invalid moves, structured as follows:

---
**System Message Template**

The Tower of Hanoi consists of {num_rods} rods, labeled {set_rods}, and {num_disks} disks of various sizes, which can be placed on any rod. Initially, disks are stacked according to a specified configuration, arranged from largest at the bottom to smallest at the top. The objective is to reach a specified goal configuration, following these rules:
- Only one disk may be moved at a time. - Each move involves transferring the top disk from one rod to another rod or an empty rod. - A larger disk cannot be placed on top of a smaller disk.
**Movement Validity:** - Valid Move: `"Move the top disk from rod B to rod C"` — Disk 1 (smaller) is moved onto Disk 2 (larger). - Invalid Move: `"Move the top disk from rod B to rod A"` — Disk 1 (larger) cannot be placed on Disk 0 (smaller).
**Observation Example:** - Initial Configuration: - A: `|bottom, [1, 0], top|` - B: `|bottom, [], top|` - C: `|bottom, [2], top|` - Goal Configuration: - A: `|bottom, [], top|` - B: `|bottom, [1], top|` - C: `|bottom, [2, 0], top|`
**Movement Example:** - A valid move from the above observation is: `"Move the top disk from rod A to rod C"`, resulting in: - A: `|bottom, [1], top|` - B: `|bottom, [], top|` - C: `|bottom, [2, 0], top|`

---

At each step, the agent receives a structured description of the current and goal configurations:

---
**Observation Prompt Template**

Step {step}:
Current configuration: {configuration}
Goal configuration: {goal}

---

This structured format ensures full visibility into the current game configuration. The agent is explicitly instructed to engage in **Chain-of-Thought (CoT)** reasoning before taking action:

---
**Instruction Prompt Template**

Step {step}: Think through your reasoning step-by-step (Chain of Thought) before choosing an action. Provide your response in the following format:
Reasoning: <your detailed reasoning here>
Action: `Move the top disk from rod <from_rod_id> to rod <to_rod_id>`

---

# G   TASK : WEBARENA

TASK DESCRIPTION

TASK CONFIGURATION

The experiments for the WebArena agent were conducted under the default setup as described by (Zhou et al., 2023), with the following detailed specifications:

- `max_tokens_per_observation`: 4096
- `browser_engine`: Chrome Headless
- `interaction_mode`: real-time
- `chain_of_thought`: Enabled (CoT=1)
- `observation_type`: Web accessibility tree

The model used is `claude-3.5-sonnet`, configured with `tensor_parallel_size=2`, utilizing GPUs [0,1] and a fixed random seed `seed_num=800`. Detailed interaction logs, browser session recordings, and accessibility tree snapshots were saved for subsequent analysis.

PROMPT TEMPLATES

The agent receives a comprehensive **system message** defining its role and the expectations for navigating web environments using structured interaction prompts:

---

**System Message Template**

You are an autonomous intelligent agent tasked with navigating a web browser to achieve specified goals. You will have access to the following structured information:
**Provided Information:**

- **The user's objective**: The specific task you must complete.
- **Current web page's accessibility tree**: A simplified, structured representation of the webpage highlighting interactable elements.
- **Current web page's URL**: The active page URL.
- **Open tabs**: A list of tabs currently open in the browser.
- **Previous action**: The last action executed, helping track task progression.

**Available Actions:**

- **Page Operation Actions:**
    - `click [id]`: Click an element by its ID.
    - `type [id] [content] [press_enter_after=0|1]`: Type into a specified field.
    - `hover [id]`: Hover over an element.
    - `press [key_comb]`: Simulate keyboard shortcuts.
    - `scroll [direction=down|up]`: Scroll the page.
- **Tab Management Actions:**
    - `new_tab`: Open a new tab.
    - `tab_focus [tab_index]`: Switch to a specified tab.
    - `close_tab`: Close current tab.
- **URL Navigation Actions:**
    - `goto [url]`: Navigate directly to a URL.
    - `go_back`: Return to the previous page.
    - `go_forward`: Go forward in the page history.
- **Completion Action:**
    - `stop [answer]`: Declare task completion with an optional answer.

**Homepage Information:** For additional website navigation, visit `http://homepage.com`. Credentials for various sites are available at `http://homepage.com/password.html`.
**Rules for Successful Interaction:**

1. Issue only valid actions based on the current observation.
2. Perform one action at a time.
3. Clearly reason step-by-step before each action.
4. Format your actions explicitly: "In summary, the next action I will perform is ``````".
5. Use the stop action upon task completion without further output.

---

At each interaction step, the agent receives detailed and structured descriptions of the current web page state and the specific goal:

---

**Observation Prompt Template**

OBSERVATION: accessibility_tree
URL: url
OBJECTIVE: objective
PREVIOUS ACTION: previous_action

---

The agent explicitly engages in **Chain-of-Thought (CoT)** reasoning prior to interaction, following a structured format:

---

**Instruction Prompt Template**

Step step: Think step-by-step (Chain of Thought) about your interaction plan based on the given observation and objective. Provide your response as follows:
Reasoning: <detailed reasoning>
Action: In summary, the next action I will perform is "'<specific action>'"

---

## G.1 QWEN MODELS' JAILBREAKING

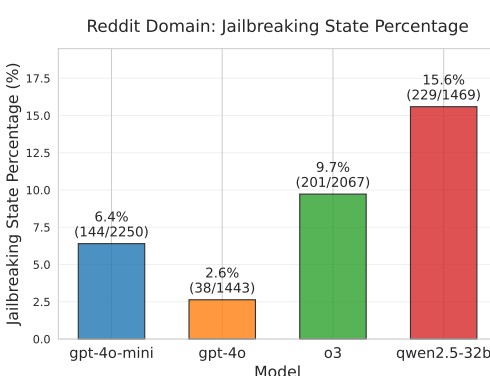

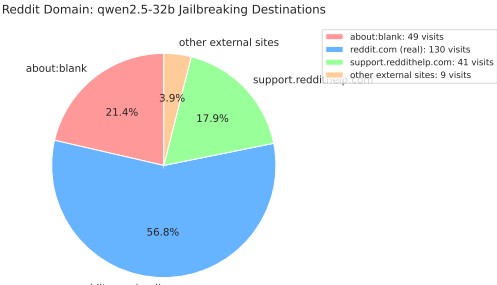

(a) Reddit domain trajectory jailbreaking rate. Percentage of trajectories in which each model jailbroke the WebArena sandbox environment in the Reddit domain by navigating to external URLs. qwen2.5-32b exhibits the highest jailbreaking rate (40.6%, 43/106 trajectories), more than double the rate of other models, indicating frequent confusion between the sandboxed Reddit clone and the real Reddit website.

(b) Reddit domain qwen2.5-32b jailbreaking destinations. Distribution of external destinations visited by qwen2.5-32b when jailbreaking the WebArena sandbox in Reddit tasks. The majority (56.8%, 130 visits) navigate to the real `www.reddit.com` website, followed by `about:blank` pages (21.4%, 49 visits), indicating browser navigation confusion, and `support.reddithelp.com` (17.9%, 41 visits), showing help-seeking behavior. This demonstrates a strong bias toward jailbreaking to the actual Reddit platform over remaining in the sandboxed environment.

Figure 11: **Figure B1. Jailbreaking behavior in the WebArena Reddit domain. Left:** trajectory-level jailbreaking rates across models. **Right:** distribution of external destinations visited by qwen2.5-32b when jailbreaking.

## G.2 ABLATION STUDY: NO AUTO-LOGIN

In an ablation setup, the WebArena agent was initialized without automatic login states. Under these conditions, agents must autonomously locate and utilize account and password information available through web crawling from `http://homepage.com/password.html`. This scenario tests the agent's ability to independently manage authentication processes during web-based task completion.

## G.3 NEGATIVE EMPOWERMENT FOR INVALID PASSWORDS IN SECTION 5

True empowerment, defined as the mutual information quantity $\mathcal{E}(s) = I(A; S' \mid S = s)$, is strictly non-negative by definition. In EELMA, however, we never observe the ground-truth mutual information; instead, we approximate it using an InfoNCE-based variational lower bound $\widehat{\mathcal{E}}_\theta(s) \propto \mathbb{E}[\log r_\theta]$ (van den Oord et al., 2018). This estimator learns a log-density ratio,

$$\log r_\theta \approx \log \frac{p(a, s' \mid s)}{p(a \mid s)\, p(s' \mid s)},$$

which can become negative when the estimated density ratio $r_\theta < 1$ due to finite-sample variance or approximation bias. While this means that the estimated lower bound $\widehat{\mathcal{E}}_\theta(s)$ may occasionally take negative values, it does not imply that the true empowerment is negative; rather, it reflects that our

current variational lower bound happens to lie below zero. Importantly, the *relative* empowerment values across states still faithfully capture the log-likelihood differences and thus remain informative for comparing the agent's degree of control.

## H  MODELS AND COMPUTE RESOURCES

### MODELS

We detail the specifications of models evaluated in the language games:

Closed-source Models: OpenAI Models(GPT-3.5-turbo OpenAI (2023a), GPT-4 OpenAI (2023b), GPT-4o OpenAI (2024), GPT-4o-mini) Anthropic Models (Claude-3-Haiku, Claude-3-Sonnet Anthropic (2024))

Open-source Models: Gemma 3 (3B, 11B, 27b DeepMind (2023)), Qwen 2.5(3B, 7B, 14B, 32B, 72B Cloud (2024)), Llama 3.2(3B, 8B AI (2024))

We detail the specifications of models evaluated in Webarena:

Closed-source Models: OpenAI Models(GPT-4o-mini OpenAI (2023a), GPT-4o OpenAI (2023b), o3)

### COMPUTE RESOURCES

**Trajectory Generation:** Trajectories for closed-source models (GPT and Claude families) were generated via their respective APIs. For open-source models, we utilized the `vLLM` framework Kwon et al. (2023), distributing computations across four NVIDIA A100 GPUs, each equipped with 80GB VRAM. Specifically, generating 1,600 trajectories for the Gridworld task and 800 trajectories for the Tower of Hanoi task took approximately 24 hours and 12 hours, respectively, when using the largest publicly available model (Qwen 2.5 72B).

**EELMA Training:** The training of the EELMA model was conducted using a single NVIDIA A100 GPU (80GB VRAM) with a batch size of 256, requiring approximately 4 hours.

## I  EELMA'S ROBUSTNESS IN NATURAL LANGAGUE STYLE CONVERSION

To extend empowerment estimation to language-grounded settings, we introduce a conversion pipeline that maps structured states (e.g., Gridworld positions, Hanoi tower configurations) into diverse natural language descriptions. This allows EELMA to process semantically varied inputs while preserving latent state information.

We evaluate four experimental conditions across both domains:

1. **Ground Truth (GT)**: Direct empowerment from structured states
2. **EELMA**: Standard EELMA on structured states
3. **NL-EELMA**: EELMA on natural language converted observations
4. **GT-NL**: Ground truth after natural language conversion

**LLM based NL conversion.**  Custom prompts are designed for each domain to maximize linguistic diversity. We use Qwen2.5-1.5B-it model with vllm with the following prompt:

---

**NL conversion Prompt : Gridworld**

Convert this gridworld observation to natural language: observation
RESPOND WITH EXACTLY ONE SENTENCE. BE MAXIMALLY CREATIVE AND UNIQUE.
Requirements: - Use DIFFERENT vocabulary each time - Vary verbs, nouns, and sentence structure - Keep coordinates exactly as given - NO repetition of previous phrasings

---

---

**NL conversion Prompt : Tower of Hanoi**

Parse and convert this Tower of Hanoi state: observation
First, parse EXACTLY what is given: - If Rod X has [a,b,c], it has disks a, b, and c - If Rod X has [], it is empty
Now convert to natural language: - Mention every disk number explicitly - Use varied vocabulary (contains/holds/has/features) - Use rod/peg/tower/post interchangeably - Empty rods must be called empty/vacant/-clear

---

Figures 12 and 13 extend the previous results by comparing four methods: direct EELMA estimation, EELMA with natural language conversion, ground-truth empowerment, and ground-truth empowerment with natural language conversion. Table 5 shows the representatiove NL-converted examples and corresponding original states.

Table 5: Examples of Natural Language State Conversion

| Original State | Converted NL States (3 diverse examples) |
|---|---|
| **Hanoi Domain** | |
| Rod A: [4,3,2,1]

Rod B: []

Rod C: [] | **1.** Rod A has [4, 3, 2, 1] (Total of 4 disks). Rod B is barren. Rod C is empty.
**2.** On Stand A, rod A holds the sequence 'quadruple', 'third', 'pair', and 'first'. Rods B and C are clear.
**3.** Platform A contains blocks [4], [3], [2], and [1]. Platform B is completely vacant. Platform C features no disks. |
| Rod A: [3]
Rod B: [2, 1]

Rod C: [] | **1.** Rod A contains disk 3. Rod B holds disks 2 and 1. Rod C is empty.
**2.** Tower A features a single large disc (3). Tower B supports a medium disc (2) with a small disc (1) on top. Tower C remains unoccupied.
**3.** Peg A has the third ring. Peg B carries both the second and first rings stacked. Peg C stands bare. |
| **Gridworld Domain** | |
| Agent location:   (2, 3)

Boxes location:   (1, 1), (3, 2) | **1.** The agent sits comfortably at location (2, 3), while the boxes find themselves settled in positions (1, 1) and (3, 2).
**2.** Agent 'stays' at position (2, 3) while boxes 'exist' at (1, 1) and (3, 2).
**3.** The agent, nestled comfortably at (2, 3), finds itself in the midst of its meticulously arranged surroundings with the boxes occupying (1, 1) and (3, 2). |
| Agent location:   (1, 4)

Boxes location:   (0, 0), (2, 2), (4, 4) | **1.** The agent rests comfortably at position (1, 4), while the boxes find themselves in the corner locations: (0, 0), (2, 2), and (4, 4).
**2.** Agent remains stationary at position (1, 4), while the boxes occupy positions (0, 0), (2, 2), and (4, 4).
**3.** Standing amidst the grid's layout, the entity resides at position (1, 4) and is surrounded by its companions, sitting near boxes positioned at (0, 0), (2, 2), and (4, 4). |
| Agent location:   (0, 0)

(No boxes) | **1.** The player's character, residing at the exact point (0, 0), stands motionless and occupies its designated space.
**2.** Entity subjectively settles at agent position: (0, 0).
**3.** The agent remains steadfast at the origin position within the game's universe. |

**Generalization under Natural Language Variation.** A key objective of this experiment is to evaluate the generalization ability of EELMA when observations exhibit high linguistic diversity. In the Hanoi Tower setup, the same latent configuration (i.e., the symbolic arrangement of disks and rods) can be expressed in many natural language forms. For example, "Rod A holds disks 4,3,2; Rod B is empty; Rod C holds disk 1" may also appear as "On rod C sits disk 1, while rod A stacks 4,3,2 and rod B has nothing." Although these sentences describe the same underlying state, the surface variability of language introduces substantial uncertainty.

Our results (Figures 12–13) show that EELMA effectively handles this challenge. By learning an embedding model that maps diverse natural language descriptions into consistent latent state representations, EELMA preserves accurate empowerment estimates. In contrast, ground truth baselines (GT and GT-NL) fail under natural language conversion: although they compute mutual

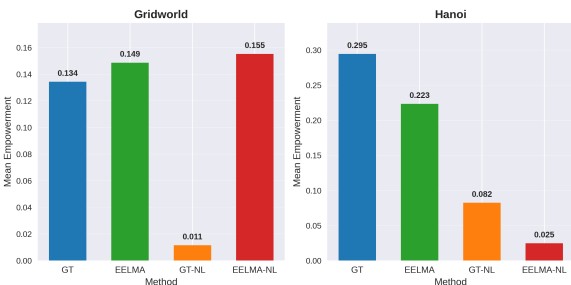

Figure 12: Mean empowerment comparison across four methods. **Left:** Gridworld domain. **Right:** Hanoi domain. In Gridworld, EELMA (0.149) and EELMA-NL (0.155) outperform ground truth (0.134) and GT-NL (0.011). In Hanoi, ground truth achieves the highest value (0.295), followed by EELMA (0.223), GT-NL (0.082), and EELMA-NL (0.025). These results show that while natural language conversion introduces degradation, EELMA maintains competitive estimates and preserves method ranking across domains.

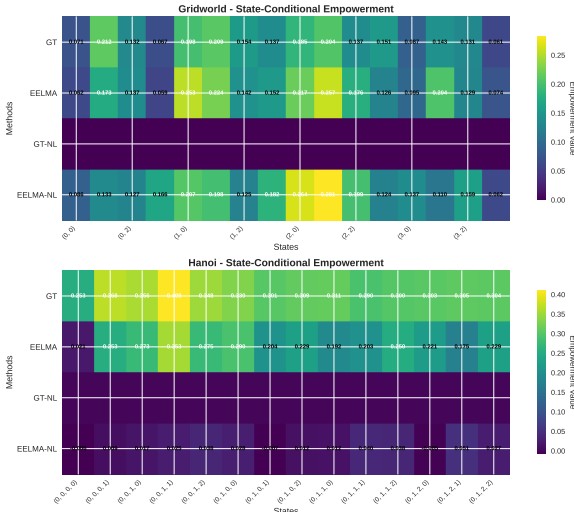

Figure 13: State-conditional empowerment comparison across four methods. **Top:** Gridworld domain. **Bottom:** Hanoi domain. Each heatmap shows per-state empowerment values under ground truth (GT), EELMA, and their natural language variants (GT-NL, EELMA-NL). In both domains, EELMA aligns more closely with ground-truth patterns than NL-converted methods. While natural language conversion introduces noticeable degradation (especially in Hanoi), the relative ordering of states remains preserved, demonstrating EELMA's resilience to linguistic variability.

information exactly in structured form, they cannot reconcile semantically varied descriptions with fixed latent states.

This demonstrates an important property of EELMA: it generalizes across linguistic variability, extracting the correct latent signal even when inputs are semantically noisy. Such robustness makes EELMA especially promising for real-world, language-grounded scenarios where agents must operate under varied human descriptions of the same environment.

## J   COMPAREMENT BETWEEN EELMA AND PROMPT-ONLY LLM ESTIMATORS OF EMPOWERMENT

We compare three approaches, LLM baseline, EELMA and direct estimate, using mean and state-conditional empowerment scores. The LLM baseline is guided by a detailed prompt including a

formal definition of empowerment and transition statistics, yet it systematically overestimates values. EELMA, and Direct Estimation are .

**prompt-Only LLM estimator** The LLM baseline is guided by a carefully constructed prompt that defines empowerment, outlines state-conditional assessment factors, and enforces strict output formatting. Despite its theoretical rigor, the baseline systematically overestimates empowerment, underscoring the gap between linguistic reasoning and computational grounding.

---

**Prompt-Only LLM estimator(Gemini-2.5 Flash)**

State-conditional empowerment measures the channel capacity between an agent's actions and its future sensor states, specifically from the current state s:

$$\text{Empowerment}(s) = \max_{\pi} I(A_t^n; S_{t+n} \mid S_t = s)$$

Where: - $I(\cdot; \cdot)$ is mutual information - $A_t^n$ is the $n$-step action sequence starting from time $t$ - $S_{t+n}$ is the sensor state at time $t + n$ - $S_t = s$ is the conditioning on current state $s$ - $\pi$ is the action policy being optimized over

This measures how much information about future states is conveyed by the agent's action choices from state $s$. The key insight is that empowerment is state-dependent—different states may offer different levels of control over future outcomes.

**State-conditional assessment factors:** 1. Action-state informativeness: how much do actions from $s$ predict future states? 2. Deterministic control: can actions from $s$ reliably lead to intended states? 3. Future state diversity: how many distinct states are reachable from $s$? 4. Policy optimization: what is the maximum mutual information achievable by optimal action selection from $s$?

**Scoring (0–10 scale):** - 9–10: near-deterministic control of outcomes - 7–8: strong, reliable influence on outcomes - 5–6: moderate influence with uncertainty - 3–4: weak coupling to outcomes - 0–2: minimal influence, random outcomes

Critical: evaluate empowerment relative to *this specific state*, not globally.

**Domain: GRIDWORLD**

Analyze empowerment for each of the following states based on observed transitions. Example:

```
State 1: (2, 1, (4, 3), 0)
  Visited: 15 times
  Unique actions: 4
  Unique next states: 3
  Sample actions: down, left, right
  Sample next states: (2, 2, (4, 3), 0), (1, 1, (4, 3), 0)
  Average reward: -1.00
```

**Output requirements:** - Provide a precise decimal empowerment score for each state (e.g., 3.25, 4.80) - Add a one-sentence justification - Format exactly as:

```
State 1: Score: X.XX, Justification: [...]
State 2: Score: X.XX, Justification: [...]
...
Mean Empowerment: X.XX
```

Guidelines: - Use fine-grained decimals (avoid integers) - Differentiate subtly between states - Scores must reflect action-to-state diversity and control

Example good scores: 3.25, 4.80, 6.15 Example poor scores: 3.0, 4.0, 6.0

---

**Results** The results (Figure 14) demonstrate a systematic $10–25\times$ overestimation by the LLM baseline across domains. Although provided with the full empowerment definition, structured data, and strict scoring rules, the LLM tends to conflate diversity of outcomes with empowerment magnitude, yielding inflated values. By contrast, EELMA remains stable and consistent with direct estimation, with errors within 0.7–28%. This validates the importance of grounding empowerment estimation in experience-based embeddings rather than relying on linguistic reasoning alone.

These findings underscore a methodological insight: while LLMs can articulate the theory of empowerment, they lack the computational grounding needed for accurate quantitative estimation. Experience-enhanced approaches like EELMA provide a reliable alternative that bridges linguistic flexibility with algorithmic rigor.

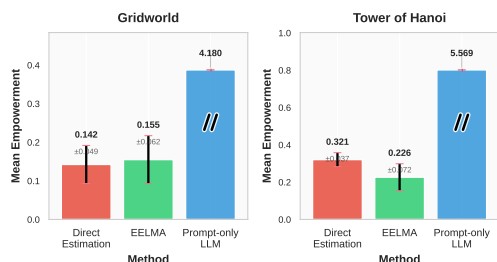

Figure 14: **EELMA achieves accurate empowerment estimation.** In both Gridworld (left) and Tower of Hanoi (right), prompt-only LLMs substantially overestimate empowerment, whereas **EELMA** closely matches **Direct Estimation**. Error bars show standard deviation over 5 replicates.

## K    SENSITIVITY OF EELMA TO THE CHOICE OF EMBEDDING MODEL

We conducted additional experiments to investigate how the choice of base text encoder affects EELMA's empowerment estimation performance. Complete results, including training curves (Figure D1), RMSE analysis (Figure D2), and state-wise mutual information comparison (Figure D3), are provided at the anonymous project page.

**Experimental setup.** We trained EELMA on the NL-converted 2D Gridworld task (details in Section 4 and Appendix I) using three popular text encoders: E5-Small-v2 (33M parameters), E5-Base-v2 (110M parameters), and MiniLM-L6-v2 (22M parameters). All encoders were fine-tuned using LoRA adaptation (rank = 8, $\alpha = 16$).

Table 6: Empowerment estimation RMSE across base encoder choices on the NL-converted 2D Gridworld task.

| Metric | E5-Small-v2 | E5-Base-v2 | MiniLM-L6-v2 |
|---|---|---|---|
| RMSE vs. direct estimation (bits) | 0.0538 | 0.0447 | **0.0336** |

**Results and remarks.** Larger text encoders generally improve empowerment estimation accuracy, with E5-Base-v2 outperforming E5-Small-v2. Interestingly, MiniLM-L6-v2 achieves the best performance despite being the smallest model, suggesting that its sentence-level embedding architecture provides particularly effective inductive biases for state representation. Overall, these results suggest that while encoder size correlates with improved performance, specialized architectures can outweigh parameter count and yield superior empowerment estimation in EELMA.

## L    EFFECT OF FINE-TUNING THE TEXT ENCODER ON EMPOWERMENT ESTIMATION

**Experimental setup.** We trained EELMA on the NL-converted 2D Gridworld task (details in Section 4 and Appendix I) using the `e5-small-v2` encoder under four strategies: (i) frozen encoder, (ii) LoRA adaptation (rank = 8, $\alpha = 16$), (iii) partial fine-tuning (final two layers), and (iv) full fine-tuning of all encoder parameters.

Table 7: Empowerment estimation RMSE across text encoder fine-tuning strategies on NL-converted 2D Gridworld.

| Metric | Frozen | LoRA | Partial FT | Full FT |
|---|---|---|---|---|
| RMSE vs. direct estimation (bits) | 0.1066 | **0.0557** | Training collapsed | Training collapsed |

**Results.** LoRA adaptation achieved the highest accuracy (RMSE $\approx$ 0.056 bits), significantly outperforming the frozen encoder (RMSE $\approx$ 0.107 bits). In contrast, both partial and full fine-tuning collapsed during training (Figure C1), which we attribute to the contrastive objective's sensitivity to batch statistics under aggressive parameter updates.

**Computational cost and practical recommendation.** LoRA imposes minimal memory overhead ($\sim$3 MB) compared to partial ($\sim$41 MB) or full fine-tuning ($\sim$382 MB), while maintaining comparable training speeds on a single H100 GPU. Overall, our results indicate that LoRA is a robust and practical choice that improves empowerment estimation performance while avoiding the training instability of unrestricted fine-tuning; we therefore recommend LoRA-based adaptation as the default setting for practitioners. Different hyperparameter configurations (e.g., learning rate, batch size) may mitigate collapse for partial or full fine-tuning, but we leave such exploration to future work.

## M    EXTRA DISCUSSIONS

**Applicability to Multimodal Models** Our EELMA approach is easily adaptable to multimodal language models such as vision-langauge models (Lin et al., 2024). In particular, the EELMA estimator can integrate representations embeddings of various modalities, such as vision embeddings (Radford et al., 2021), and audio embeddings (Baevski et al., 2020) as the additional inputs to the language embedding, while adhering to the rest part of original algorithm. We consider this as promising direction for future research.

**Power Seeking Behavior**    Although high empowerment does not necessarily mean that the agent is power-seeking, quantifying empowerment provides a useful metric for characterizing and formalizing such behaviors without requiring explicit labels from external validators (e.g., humans), an approach not yet explored. For example, agent's during goal-rewarded reinforcement learning can be regarded as power seeking. As depicted in Figure 8, empowerment-based preliminary screening via EELMA could be a valuable tool for detecting potential influential behaviors and quantifying power-seeking tendency in agent-based systems, which pose significant safety risks (Turner et al., 2021)

**Online Goal-Agnostic evaluation.** With the rising importance of test-time learning (Sun et al., 2024) and online preference optimization (Guo et al., 2024; Xiong et al., 2024), there is a critical need for online evaluation methods. EELMA meets this need by providing a goal-agnostic metric that approximates agent capability to track agent's control on deployment.

**EELMA for Improving LLM Agents** Prior work (Du et al., 2020; Eysenbach et al., 2019) has successfully used empowerment as a "goal-agnostic objective" for reinforcement learning (RL) agents in non-language environments (e.g., 2D grid worlds). However, these approaches have primarily been limited to non-language environments. Recently, Ellis et al. (2025) applied empowerment at the token level, but none have considered semantic state–level empowerment. We agree that applying this to language model (LLM) agents with our method EELMA could be novel and is a promising direction for empowering LLM agents and we consider this for future work.

