# OpenReview forum: "Estimating the Empowerment of Language Model Agents"
_ICLR.cc/2026/Conference — Submitted to ICLR 2026_

### Official Review · Reviewer_md1k · 2025-10-29

**Soundness:** 3
**Presentation:** 3
**Contribution:** 3
**Rating:** 8
**Confidence:** 3

**Summary:**

This paper proposes an information-theoretic framework, called EELMA (Empowerment Estimation for Language Model Agents), to quantify the empowerment of LLM-based agents. Empowerment is defined as the mutual information between an agent’s actions and the resulting future states, representing how much the agent can influence its environment. EELMA applies this concept to language-based settings, where actions and states are represented by text embeddings, and mutual information is estimated via contrastive learning (InfoNCE). The method is evaluated in several environments, including Gridworld, Tower of Hanoi and WebArena. Results show that empowerment correlates strongly with task reward and captures patterns of agentic behavior such as controllability and power-seeking.

**Strengths:**

1. Timely and relevant topic.

The paper addresses a central question for the emerging field of LLM-based agents: how to measure agentic capability in a goal-free, interpretable, and scalable way.

2. Conceptual novelty.

Adapting empowerment from control theory to language environments is creative. It introduces an information-theoretic lens to quantify an agent’s potential influence without relying on pre-defined tasks or rewards.

3. Goal-agnostic evaluation framework.

The work moves beyond conventional benchmarks and offers an interesting direction for measuring emergent autonomy in complex environments.

4. Potential broader impact.

The strong correlation between empowerment and actual performance across diverse settings suggests that empowerment captures meaningful structure in agent behavior. Empowerment could become a useful signal for monitoring or constraining power-seeking tendencies, which connects to safety and interpretability concerns in agentic AI.

**Weaknesses:**

1. Conceptual ambiguity in defining empowerment for language models.

In its original formulation, empowerment measures how an agent’s actions causally influence future physical states. In language models, however, the “future state” is a symbolic context, not a physical one. This makes the notion of empowerment somewhat ambiguous: the estimated mutual information may reflect linguistic diversity or discourse influence rather than true causal control.

2. Simplified experimental environments.

The toy setups (Gridworld, Tower of Hanoi) are helpful for clarity but limited in linguistic and social complexity. While the WebArena tests are more realistic, the evaluation still relies on scripted interactions. The claim of scalability would be stronger with more open-ended or multi-agent scenarios (Maybe Minecraft?).

**Questions:**

* Do you envision empowerment being used not only as an evaluation metric but also as a design for improving LLM agents? For example, could empowerment be incorporated into agent training or architecture design?

---

> ### Author Response · Authors · 2025-11-24
>
> Thank you for your thoughtful and encouraging review. We are particularly encouraged by your recognition that our design is "principled and clear," with a schema-agnostic structure that makes transferability inherent. We also appreciate your praise that the RL approach is well-suited for multi-turn KG interaction and that the results showcase "both efficiency and accuracy." In response to your thoughtful review, we have added new experimental results via an Anonymous link: https://sites.google.com/view/anonymouseelma.
>
> ---
> **1. Ambiguity in empowerment definition for langauge models.**
>
> We clarify that our work focuses on **empowerment over symbolic state**, i.e control over the underlying state configuration rather than surface-level observations.
> - **Empowerment over underlying symbolic state** : In language environments, observations often exhibit a one-to-many mapping where a single symbolic state (e.g., grid position (1,2) in a 2D Gridworld) maps to multiple textual descriptions (e.g., "the agent is at (1,2)" vs. "agent is located at x=1, y=2"). Consequently, a valid empowerment metric must be invariant to these phrasing differences to capture the true degree of control that the agent exerts. EELMA addresses this issue by estimating empowerment over a latent embedding space that clusters semantically equivalent descriptions. This ensures the metric reflects the agent's actual control over semantic state transitions, invariant to surface-level linguistic variability (Section 3.4).
>
> We will explicitly clarify this distinction and its necessity in the updated manuscript.
>
> ---
> **2. Why choosing simplified environment?**
>
> We deliberately selected language games like Gridworld and Tower of Hanoi as controlled environments to validate EELMA’s precision and generalization to natural language variability. This validation was a prerequisite for applying the method to the more complex, realistic WebArena environment.
>
> ---
> **3. Scalability to more open-ended or multi-agent scenarios.**
>
> We agree that truly open-ended environments (e.g., Minecraft[1], OSWorld [2], GAIA [3]), multi-agent scenarios multi-agent scenarios are important for future generalization. Additionally, we propose extending our framework toward multimodal interactions, as discussed in Appendix K. We recognize the importance of testing our method in these broader contexts and view it as a meaningful direction for future work.
>
> ---
> **4. Can empowerment used for improving LLM agents?**
>
> Yes, empowerment can be incorporated into agent training. Prior work has successfully used empowerment as a “goal-agnostic objective” for reinforcement learning (RL) agents in non-language environments (e.g., 2D grid worlds), such as AvE, DIAYN, ESR [4,5,6]. However, these approaches have primarily been limited to non-language environments. Recently, the arXiv paper [7] applied empowerment at the token level, but none have considered semantic state–level empowerment. We agree that applying this to language model (LLM) agents with our method EELMA could be novel and is a promising direction for empowering LLM agents and we consider this for future work.
>
> References
>
> - [1] Minecraft : Wang, G., et al. (2024). *Voyager: An Open-Ended Embodied Agent with Large Language Models*. Transactions on Machine Learning Research.
> - [2] OSWorld: Xie, T., et al. (2024). *OSWorld: Benchmarking Multimodal Agents for Open-Ended Tasks in Real Computer Environments*. NeurIPS 2024..
> - [3] GAIA: Mialon, G., et al. (2024). *GAIA: A Benchmark for General AI Assistants*. ICLR 2024.
> - [4] AvE: Du, Y., et al. (2020). *AvE: Assistance via Empowerment*. NeurIPS 2020.
> - [5] DIAYN : Eysenbach, B., et al. (2019). *Diversity is All You Need: Learning Skills without a Reward Function*. ICLR 2019.
> - [6] ESR : Myers, V., et al. (2025). *Learning to Assist Humans without Inferring Rewards*. NeurIPS 2024.
> - [7] Ellis, Evan, et al. "Training LLM Agents to Empower Humans." *arXiv preprint arXiv:2510.13709* (2025).

---

### Official Review · Reviewer_5sQU · 2025-10-31

**Soundness:** 2
**Presentation:** 3
**Contribution:** 2
**Rating:** 4
**Confidence:** 4

**Summary:**

The paper introduces a method for evaluating the capabilities of LM-based agents based on empowerment. They propose EELMA, a way to estimate empowerment from multi-turn text interactions using existing approaches to contrastive (InfoNCE) mutual-information estimation combined with text embedders. Results show that across controlled language-based environments like Gridworld, Tower of Hanoi and web-browsing tasks (WebArena), EELMA’s empowerment estimates strongly correlate with average task performance, decrease with environmental complexity, and increase with model scale, memory, and chain-of-thought reasoning. Other empirical results suggest that EELMA can flag pivotal actions such as authentication events, pointing towards use cases for open-ended evaluation or safety monitoring of LM agents.

**Strengths:**

- As far as I can tell, this paper is the first to investigate measuring empowerment in text-based domains.

- The paper’s experiments convincingly demonstrate that the EELMA estimate is close to the direct empowerment in the Gridworld and Tower of Hanoi environments.

- The experiments in the Gridworld and Tower of Hanoi are quite rigorous and explore scaling capabilities across several axes (model size, CoT, memory, etc.). The results show convincing trends across several model types.

- The paper is easy to understand and read and the graphics cleanly present experiment results.

**Weaknesses:**

- The algorithmic contribution is minor, combining existing mutual information estimation approaches for empowerment with text embedders so that they can be used with LM agents in text domains.

- Existing papers have demonstrated the same result of the main claim (that effective empowerment can approximate agentic performance) in environments very similar to Gridworld and Tower of Hanoi, simply without representing the environment and actions in text. The experiments in this domain are really only showing me that adding the text-based embedding on top of infoNCE doesn’t alter the results of existing works.

- The experiments in the more complicated domain of WebArena are more limited (i.e., only 3 data points across each type of environment) and demonstrate mixed results. I think putting a larger emphasis on novel domains (e.g. WebArena) instead of environments in which empowerment has been thoroughly explored (gridworlds or similar small, discrete games) would greatly strengthen the paper.

- Training the embedders and successor representation requires additional rollouts (potentially synthetically generated as mentioned in the paper) which limits the generality of the method. Are there any ideas for relaxing the need to collect additional rollout in the specific environment so that the measure of empowerment can generalize to unseen environments?

**Questions:**

- Why was a p value of 0.001 chosen in the right side of Figure 8? This makes the error bars overlap and doesn’t give a meaningful sense of the statistical significance. I would recommend choosing a p-value that makes the error bars not overlap.

- Your estimated empowerment for invalid passwords in section 5 is negative. Shouldn’t empowerment always be non-negative? What is going on here?

- How do you expect your approach and results to scale to more complicated and longer horizon tasks? What is the current main bottleneck to scaling up?

- Why did you choose to use discounted rewards in WebArena instead of success rate (as far as I know, the original WebArena work uses success rate)? How do the results change with success rate instead of discounted reward?

Misc:
- Appendix K has some formatting errors
- Line 1261 and 1402 has an undefined ref
- Section C.1 has several punctuation errors and typos.

---

> ### Author Response · Authors · 2025-11-24
>
> Thank you for your thoughtful and insightful review. We are encouraged by your recognition that this paper is "the first to investigate measuring empowerment in text-based domains." We also deeply appreciate your observation that our experiments are "quite rigorous" and "convincingly demonstrate" the validity of the EELMA estimate. In response to your questions about additional WebArena data and the statistical tests for Figure 8, we have added new experimental results via an Anonymous link: https://sites.google.com/view/anonymouseelma.
>
> ---
> **1. Novelty.**
>
> Our contributions lie in the first application of empowerment at the scale of Large Language Models (LLMs) and our solution to the unique challenges of text-based state spaces. We summarize these novelties as follows:
>
> - **First empowerment estimator for text environments:** While Mutual Information (MI) estimation is established in robotics, applying it to the high-dimensional, sparse, and semantic nature of language is non-trivial. EELMA is the first framework to address challenges such as linguistic variability and uncertainty by effectively mapping textual observations to compact embeddings for variational estimation.
> - **Formalization of empowerment as goal-agnostic evaluation metric:** We formalize effective empowerment as a scalable, goal-agnostic metric that complements or replaces costly conventional benchmarks. We empirically demonstrate that our estimates strongly correlate with agentic performance (measured via discounted reward) across diverse tasks.
> - **Comprehensive case study**: We apply EELMA to analyze how specific components (CoT, model scale, memory) and environmental complexity modulate an agent's power. This provides a comprehensive diagnosis of the factors that empower agents and drive agentic performance.
> - **Proposal for safety monitoring:** We propose and demonstrate EELMA's capacity to detect highly influential action states (e.g., authentication) solely through empowerment spikes, enabling safety monitoring without reliance on explicit reward signals.
>
> ---
> **2. Main hypothesis and choice of environments.**
>
> - We acknowledge that 2D Gridworld and Tower of Hanoi have been studied in non-text contexts (e.g., Myers et al., 2025 [1] or Du, Y., et al. 2020 [2]). However, we selected them intentionally as controlled environments to establish and validate EELMA’s reliability before applying it to more realistic domains like WebArena.
>
> - Also, we clearly state that our main hypothesis *"empowerment proxy the average performances"* are motivated by the previous works [1][2][3] and cited. However, our distinct novelty is that we are the first to empirically validate this hypothesis for LLM agents in natural language spaces with proposed methodology EELMA.
>
> ---
>
> **3. Limited datapoints in Webarena.**
>
> We agree on the need for broader experimental analysis, so we evaluated **Qwen 2.5 32B** on WebArena, keeping all other settings identical.
>
> - **The trend is robust**: As shown in Figure A (see anonymous link), we observe a strong positive correlation between empowerment and discounted reward across the Github, Reddit, and Shopping Admin domains, further substantiating our main hypothesis. In the Shopping domain, however, the correlation remained flat even with the addition of Qwen 2.5. This result is consistent with the limitations discussed in our manuscript, that tasks heavily dependent on semantic reasoning (answering specific price values) may not align with the ability to state control that empowerment metric quantifies.
>
>
> - **Spurious empowerment from Qwen's jailbreaking (Reddit)**: Interestingly, Qwen performs poorly on Reddit tasks yet maintains quite high empowerment comparable to GPT-4o-mini and o3 (Figure A), showing a non-negligible offset from the linear fitting line. We found this is due to 'jailbreaking' behavior: 40% of Qwen’s trajectories navigate to external websites (e.g., the real “www.reddit.com”) rather than the WebArena sandbox server, shown in Figure 2B in the anonymous link. Consequently, the model navigates to diverse but task-irrelevant external websites; this artificially inflates state diversity and empowerment estimates, even though the agent fails the task objectives.
>
> We appreciate the reviewer prompting this analysis and will include these valuable observations in the updated manuscript
>
> ---
> **4. Testing on realistic environments?**
>
> We agree that more realistic environments such as A2Perf[4] and TravelPlanner[5], and open-ended environments (e.g., Minecraft[6], OSWorld [7], GAIA [8]) are important for future generalization. Additionally, we propose extending our framework toward multimodal interactions, as discussed in Appendix K. We recognize the importance of testing our method in realistic setup and view it as a key direction for future work.

---

> ### Author Response · Authors · 2025-11-24
>
> ---
> **5.Statistical significacne in the right pannel of Figure 8.**
>
> We sincerly thank the reviewer for pointing out this error. We intended to plot sample means with Standard Error of the Mean (SEM) but inadvertently plotted sample variance instead. We have corrected this in Figure E1 (available at the anonym link).
>
> - **Statistical analysis**: To clarify, the p-value was not chosen arbitrarily but calculated from the data. Using Welch's t-test to compare valid authentication (mean = 0.2685 bits, n=702) versus invalid authentication (mean = -0.0937 bits, n=287), we obtained t(729.7) = 7.843 with corresponding p < 0.001. The overlap of SEM error bars (in the updated Figure E1) does not contradict statistical significance, as error bar overlap is not equivalent to a formal hypothesis test. We will clarify this in the revised manuscript and update the figure caption to explicitly state that error bars represent SEM.
>
>
> ---
> **6.Negative empowerment for invalid passwords in section 5.**
>
> True empowerment, defined as the mutual information quantity $\mathcal{E}(s) = I(A; S' \mid S=s)$, is indeed strictly non-negative by definition. In EELMA, however, we approximate this quantity using an InfoNCE-based variational lower bound $\widehat{\mathcal{E}}\theta(s) \propto \mathbb{E}[\log r\theta]$ [3]. This estimator learns a log-density ratio $\log r_\theta \approx \log \frac{p(a,s' \mid s)}{p(a \mid s)p(s' \mid s)}$. Unlike the theoretical MI, this learned bound can be negative when the estimated density ratio $r_\theta < 1$ due to finite-sample variance or approximation bias. Consequently, the \emph{relative} empowerment values accurately reflect the log-likelihood differences.
>
> ---
> **7. Scalability & Bottleneck to more complicated and longer horizon tasks.**
>
> The computational cost of trajectory acquisition can indeed be a challenge for scaling. Intrinsically, estimating empowerment requires sampling a sufficient number of trajectories to approximate the state-action distribution. In the worst-case scenario, assuming a dense transition MDP environment where all states are reachable, the number of possible trajectories scales exponentially with the horizon $T$, roughly $O(|A|^T)$, where $|A|$ is the action space size. We will add a discussion on this ``curse of dimensionality'' to the Limitations section.
>
> However, we argue that this theoretical complexity does not prohibit scaling to real-world tasks. In many practical scenarios, the effective branching factor is significantly smaller than the theoretical action space. Although an agent \textit{can} take many actions, only a sparse subset of trajectories leads to meaningful state transitions or non-zero rewards. Our method leverages this sparsity, allowing efficient estimation even in longer-horizon tasks without exploring the entire exponential trajectory space.
>
> ---
> **8. Why choose discounted rewards over success rate for WebArena?**
>
> We selected discounted rewards to align with the theoretical definition of empowerment. As Section 3 establishes that effective empowerment (incorporating discount factor $\gamma$) serves as a lower bound for average discounted reward, this metric allows us to verify our theoretical predictions more directly than binary success.
>
> ---
> References
>
> - [1] ESR : Myers, V., et al. (2025). *Learning to Assist Humans without Inferring Rewards*. NeurIPS 2024.
> - [2] AvE: Du, Y., et al. (2020). *AvE: Assistance via Empowerment*. NeurIPS 2020.
> - [3] Ellis, Evan, et al. "Training LLM Agents to Empower Humans." *arXiv preprint arXiv:2510.13709* (2025).
> - [4] Uchendu, I., et al. (2025). *A2Perf: Real-World Autonomous Agents Benchmark*. arXiv:2503.03056.
> - [5] Xie, J., et al. (2024). *TravelPlanner: A Benchmark for Real-World Planning with Language Agents*. ICML. arXiv:2402.01622.
> - [6] Minecraft : Wang, G., et al. (2024). *Voyager: An Open-Ended Embodied Agent with Large Language Models*. Transactions on Machine Learning Research.
> - [7] OSWorld: Xie, T., et al. (2024). *OSWorld: Benchmarking Multimodal Agents for Open-Ended Tasks in Real Computer Environments*. NeurIPS 2024..
> - [8] GAIA: Mialon, G., et al. (2024). *GAIA: A Benchmark for General AI Assistants*. ICLR 2024.
> - [9] Oord, Aaron van den, Yazhe Li, and Oriol Vinyals. "Representation learning with contrastive predictive coding." *arXiv preprint arXiv:1807.03748* (2018).

---

### Official Review · Reviewer_Ky3e · 2025-11-03

**Soundness:** 3
**Presentation:** 3
**Contribution:** 3
**Rating:** 6
**Confidence:** 4

**Summary:**

This paper proposes EELMA, a method to estimate the empowerment of language model agents. Empowerment is defined as the mutual information between an agent’s actions and its future states. The authors argue that it provides a task-agnostic, open-ended way to measure agentic capabilities. They implement EELMA using embedding-based representation learning and InfoNCE-style mutual information estimation. The method is evaluated in both structured environments (GridWorld and Tower of Hanoi) and a more realistic setting (WebArena). Experiments show that empowerment correlates well with goal-conditioned performance, and can highlight influential decision points in trajectories.

**Strengths:**

1. Introduces a task-agnostic, theoretically grounded metric for evaluating LLM agents.

2. Proposes a scalable estimation algorithm for empowerment in language-based environments.

3. Shows empirical evidence of strong correlation between empowerment and average task reward.

**Weaknesses:**

1. The semantic meaning of empowerment remains abstract in complex language settings.

2. The proposed method still relies on known reward functions for validation, which weakens its claim of being fully task-agnostic.

3. The computational cost of training EELMA is not discussed in practical terms.

**Questions:**

1. How sensitive are the empowerment estimates to the choice of embedding model?

2. Would fine-tuning the embeddings improve performance, and at what cost?

3. How can EELMA be used in practice, for example, during online deployment?

4. In settings where high empowerment is not aligned with human goals, how should we interpret or act on those results?

---

> ### Author Response · Authors · 2025-11-24
>
> Thank you for your thoughtful and encouraging review. We are particularly gratified by your appreciation of our work's primary contributions: introducing a *"task-agnostic, theoretically grounded metric"* and proposing a *"scalable estimation algorithm"* for language-based environments. We are also glad that you found the empirical evidence demonstrating a *"strong correlation between empowerment and average task reward"* to be convincing. In response to your questions about impact of embedding models and effectiveness of finetuning, we have added new experimental results via an Anonymous link : https://sites.google.com/view/anonymouseelma.
>
> ---
> **1. The semantic meaning of empowerment remains abstract in complex language settings.**
>
> We emphasize that our work focus on **empowerment over symbolic state**, i.e control over the underlying state configuration rather than surface-level observations. To clarify :
>
> - **Meaning of Empowerment**: Generally, empowerment measures an agent's capability to control its future state. As illustrated in Figure 1, a low-empowerment agent is trapped in repetitive loops, whereas a high-empowerment agent can generate actions that access a diverse range of semantically distinct future states (e.g., navigating to an admin panel) by effectively aligning its actions with its intentions. Formally, empowerment is defined as the mutual information between action and future states.
>
> - **Empowerment over underlying symbolic state**:
> In language environments, observations often exhibit a one-to-many mapping where a single symbolic state (e.g., grid position (1,2) in a 2D Gridworld) maps to multiple textual descriptions (e.g., "the agent is at (1,2)" vs. "agent is located at x=1, y=2"). Consequently, a valid empowerment metric must be invariant to these phrasing differences to capture the true degree of control that the agent exerts. EELMA addresses this issue by estimating empowerment over a latent embedding space that clusters semantically equivalent descriptions. This ensures the metric reflects the agent's actual control over semantic state transitions, invariant to surface-level linguistic variability (Section 3.4).
>
> We will explicitly clarify this distinction and its necessity in the updated manuscript.
>
> ---
> **2. The proposed method still relies on reward functions and not fully task-agnostic.**
>
> We apologize for the lack of clarity regarding this point. To clarify:
>
> - **EELMA is a stand-alone method** that estimates empowerment solely from state-action trajectories. This means EELMA can measure empowerment without requiring reward functions or end-goal setups.
> - **Reason behind discount reward calcaultion**: Throughout the paper, we calculated discounted rewards strictly to validate the hypothesis that empowerment "empirically serves as a proxy for average goal reward" (lines 212-213). This confirms that empowerment can serve as a standalone metric for the agentic performance of language model agents.
> - **Remark**: In practice, EELMA is designed for deployment without reward calculation. This is demonstrated in our Authentication case study (Section 5): although "there are no rewards associated with either of these steps" (line 438), EELMA successfully identified valid authentication as a high-empowerment state, proving its utility as an evaluation metric even when explicit reward functions are absent. We will clarify this in the introduction of the manuscript.
>
> ---
> **3. The computational cost of training EELMA is not discussed in practical terms.**
>
> To summarize, we use a single NVIDIA A100 80GB GPU (embedding base: E5-base-v2) for Gridworld and Tower of Hanoi, and two GPUs with parallelization for WebArena (embedding base: E5-7b). This configuration allows us to execute the entire pipeline (including embedding models, InfoNCE estimators, and gradient descent with Adam optimizer) with a training batch size of 256 using bf16 precision. Training is stopped when the estimated contrastive loss converges, which occurs around 10,000 steps. This end-to-end training of EELMA takes approximately 4 hours for Gridworld and Tower of Hanoi (following the setups in Appendices E and F) and 7 hours for WebArena (following Appendix G). We will clarify these details in the Appendix.

---

> ### Author Response · Authors · 2025-11-24
>
> **4. Would fine-tuning the embeddings improve performance, and at what cost?**
>
> We conducted additional experiments to investigate the impact of fine-tuning the text encoder on EELMA's performance. Complete results, including training curves (Figure C1), RMSE analysis (Figure C2), and state-wise MI comparison (Figure C3), are available at the Anonymous link.
>
>
> **Experimental Setup:** We trained EELMA on the NL-converted 2D Gridworld task (details in Section 4 and Appendix I) using the `e5-small-v2` encoder under four strategies: Frozen encoder, LoRA adaptation (rank=8, alpha=16), Partial fine-tuning (final 2 layers), and Full fine-tuning.
>
> **Results:**
>
> Table 1: Empowerment estimation RMSE across text encoder fine-tuning strategies.
> | #Metric | #Frozen Encoder | #LoRA Adaptation | #Partial Fine-tuning | #Full Fine-tuning |
> | :--- | :---: | :---: | :---: | :---: |
> | RMSE vs. Direct Estimation (bits) | 0.1066 | **0.0557** | Training Collapsed | Training Collapsed |
> - **Performance:** LoRA adaptation achieved the highest accuracy (RMSE: 0.056 bits), significantly outperforming the frozen encoder (RMSE: 0.107 bits). Surprisingly, partial and full fine-tuning collapsed during training (Figure C1), due to the contrastive objective's sensitivity to batch statistics under aggressive parameter updates
> - **Computational Cost:** LoRA imposes minimal memory overhead (3 MB) compared to partial (41 MB) or full fine-tuning (382 MB), while maintaining comparable training speeds on a single H100 GPU.
>
> **Remarks:** Our results suggest that LoRA is a robust choice that that improves empowerment estimation performance while avoiding the training instability of unrestricted fine-tuning, recommend as the practictioner. The failure of partial and full fine-tuning likely stems from stochastic batch sampling in contrastive learning, though different hyperparameter configurations (learning rate, batch size) may yield different outcomes.
>
> ---
> **5.How sensitive are the empowerment estimates to the choice of embedding model?**
>
> We conducted additional experiments to investigate the impact of base embedder model choice on EELMA's empowerment estimation performance. Complete results, including training curves (Figure D1), RMSE analysis (Figure D2), and state-wise MI comparison (Figure D3) are available at the Anonymous link.
>
> **Experimental Setup:** We trained EELMA on the NL-converted 2D Gridworld task (details in Section 4 and Appendix I) using three popular text encoders: E5-Small-v2 (33M params), E5-Base-v2 (110M params), and MiniLM-L6-v2 (22M params). All encoders were fine-tuned using LoRA adaptation (rank=8, alpha=16).
>
> **Results:**
>
> Table 2: Empowerment estimation RMSE across base embedder model choice.
> | #Metric | #E5-Small-v2 | #E5-Base-v2 | #MiniLM-L6-v2 |
> | :--- | :---: | :---: | :---: |
> | RMSE vs. Direct Estimation (bits) | 0.0538 | 0.0447 | **0.0336** |
>
> - **Performance:** Larger text encoders generally improve empowerment estimation accuracy, with E5-Base-v2 outperforming E5-Small-v2. However, MiniLM-L6-v2 achieved the best performance despite being the smallest model, suggesting its sentence-level embedding architecture provides particularly effective inductive biases for state representation.
>
>
> **Remarks:** Our results suggests that while encoder size correlates with improved performance, specialized architectures can outweigh parameter count and have improved empowerment estiamtion performance in EELMA.
>
> ---
> **6. How can EELMA be used in practice, for example, during online deployment?**
>
> We envision two primary use cases for online deployment:
>
> - **Online goal-agnostic evaluation**: With the rising importance of test-time learning[1] and online preference optimization[2,3], there is a critical need for online evaluation methods. EELMA meets this need by providing a goal-agnostic metric that approximates agent capability to track agent's control on deployment.
>
> - **Safety monitoring**: As demonstrated in our Case Study (Figure 8), empowerment can identify highly influential states and actions, such as successful authentication. This enables the open-ended monitoring of safety risks, such as power-seeking tendencies [4,5], during live agent interactions without needing explicit labels. We have discussed this application more in Appendix K (”Extra discussions”)
>
> ---
> References
>
> - [1] Sun et. al., Learning to (Learn at Test Time): RNNs with Expressive Hidden States. arXiv:2407.04620, 2024
> - [2] Guo et. al., Direct Language Model Alignment from Online AI Feedback. arXiv:2402.04792, 2024
> - [3] Xiong et. al., Iterative Preference Learning from Human Feedback: Bridging Theory and Practice for RLHF under KL-Constraint. ICML 2024
> - [4] Turner et. al., Optimal policies tend to seek power. In Advances in Neural Information Processing Systems 34 NeurIPS, 2021
> - [5] Turner et. al., Parametrically retargetable decision-makers tend to seek power. NeurIPS, 2022

---

### Author Response · Authors · 2025-11-26
**Revision Note**

Dear PCs, SACs, ACs, and Reviewers,

Thank you for your valuable suggestions. **To address the comments raised by the reviewers, we have made the following adjustments to the paper**. Changes are highlighted in green.

- We have added a point-by-point breakdown of our novelties and core contributions in the Introduction. *(Reviewers 5sQU, wf7t)*

- We agree that the earlier manuscript could be misread as suggesting that EELMA requires explicit reward functions, which is not true. We have revised the Introduction to explicitly characterize EELMA as a standalone framework that does not rely on task-specific reward functions. *(Reviewer Ky3e03)*

- We have revised Section 3 to explain why discounted reward, rather than success rate, is coupled with future state reachability (in empowerment) and how this relates to cumulative return. *(Reviewers 5sQU, wf7t)*

- We have revised Section 3 to elucidate the semantic meaning of empowerment in this context, highlighting the unique challenges of estimation in language spaces. Crucially, we emphasize that our definition operates on symbolic states rather than surface-level textual observations, ensuring robustness to phrasing variations. *(Reviewers Ky3e03, 5sQU, wf7t)*

- We have added detailed ablation studies on the choice of base encoders and the degree of fine-tuning in Appendices K and L, and summarized them in a new “Implementation Guidelines” subsection in Section 4. *(Reviewer Ky3e03)*

- We have included new evaluation results for Qwen2.5-32B on the WebArena environment (Figure 7), along with an observation of frequent jailbreaking behaviors of the Qwen model. *(Reviewer 5sQU)*

- We clarified in Section 4 that, although true empowerment is by definition non-negative, our method estimates an InfoNCE-based variational lower bound that can take negative values due to finite-sample noise and approximation error. *(Reviewer 5sQU)*

- We revised the statistical analysis of the authentication actions by using Welch’s t-test and reporting the standard error of the mean (SEM). (Figure 8) *(Reviewer 5sQU)*

- We added a discussion of scalability to more complex environments to the limitations in the Discussion section (Section 6). *(Reviewer wf7t)*

- We added extra discussions (Section M) on online deployment, generalization to unseen environments, and using empowerment to improve LLM agents. *(Reviewers Ky3e03, 5sQU, wf7t)*


We sincerely appreciate the time and consideration you devoted to reviewing our manuscript. Your thoughtful feedback has substantially strengthened the paper.


Sincerely,
The Authors

---

### Meta-Review · Area_Chair_A6Fb · 2026-01-06

**Summary:**

Reviewers generally viewed the paper as well written and addressing an important problem, and several acknowledged the appeal of introducing empowerment as a task-agnostic metric for language-model agents. However, the main concern about novelty was raised explicitly by Reviewer 5sQU, who argued that the algorithmic contribution is limited, as EELMA primarily combines existing mutual-information (InfoNCE) estimators with text embedding models. The reviewer also noted that similar conclusions, namely that empowerment correlates with agent performance, have already been demonstrated in prior work using non-text environments. While the application to language is new in scope, the core technical ideas do not substantially go beyond existing methods.

In addition, the experimental evaluation was not sufficiently compelling to support acceptance. Much of the validation relies on Gridworld and Tower of Hanoi, environments that have already been extensively explored in prior empowerment research, which limits the incremental insight provided here. Although the WebArena experiments move toward more realistic agentic settings, they are limited in scale and depth, making it difficult to assess robustness and generality. Overall, while the work has clear potential and could become impactful with stronger novelty and broader evaluation, I agree that it currently has limited novelty, relies heavily on previously studied environments, and provides only limited experimental evidence in realistic agentic environments, which motivates a recommendation to reject.

**Reviewer Concerns:**

Concerns that were addressed by the rebuttal:
The authors provided thorough clarifications regarding the semantic interpretation of empowerment in language settings, emphasizing that EELMA operates over latent symbolic states rather than surface text, which partially addresses concerns raised by Reviewers Ky3e and md1k about conceptual ambiguity. They also responded convincingly to questions about sensitivity to embedding choice and fine-tuning strategies by adding ablation studies and practical guidelines. Statistical issues and presentation errors raised by Reviewer 5sQU, such as negative empowerment values and error bar reporting, were adequately explained and corrected. Additionally, the rebuttal clarified the computational setup and training cost, which addressed practical questions about feasibility.

Concerns that remain outstanding:
The main concern raised by Reviewer 5sQU regarding limited novelty was not fundamentally resolved. While the rebuttal better articulates the motivation and positioning of the work, it does not change the fact that the core method largely combines existing mutual-information estimation techniques with text embeddings, without introducing a new algorithmic insight. Similarly, the reliance on Gridworld and Tower of Hanoi remains a limitation, as these environments have already been extensively explored in prior empowerment work, and the additional WebArena experiments, though helpful, are still limited in scale and depth. As a result, the central concern that the paper provides insufficient incremental contribution and limited experimental evidence in complex, agentic environments remains unresolved.

**Reviewer Scores:**

Reviewer Ky3e -> maintains the score as concerns were partially addressed
Reviewer 5sQU -> maintains the score as concerns were partially addressed
Reviewer md1k -> lowers the score since the reviewer weren't familiar with previous works on empowerment

---

### Decision · Program_Chairs · 2026-01-26

Reject